# Source sector and fuel contributions to ambient PM$_{2.5}$ and attributable mortality across multiple spatial scales

Erin E. McDuffie [1,2✉], Randall V. Martin [1,2], Joseph V. Spadaro[3], Richard Burnett[4], Steven J. Smith [5], Patrick O'Rourke[5], Melanie S. Hammer[1,2], Aaron van Donkelaar[2,1], Liam Bindle [1,2], Viral Shah[6,10], Lyatt Jaeglé[6], Gan Luo[7], Fangqun Yu [7], Jamiu A. Adeniran[8], Jintai Lin [8] & Michael Brauer [4,9]

Ambient fine particulate matter (PM$_{2.5}$) is the world's leading environmental health risk factor. Reducing the PM$_{2.5}$ disease burden requires specific strategies that target dominant sources across multiple spatial scales. We provide a contemporary and comprehensive evaluation of sector- and fuel-specific contributions to this disease burden across 21 regions, 204 countries, and 200 sub-national areas by integrating 24 global atmospheric chemistry-transport model sensitivity simulations, high-resolution satellite-derived PM$_{2.5}$ exposure estimates, and disease-specific concentration response relationships. Globally, 1.05 (95% Confidence Interval: 0.74–1.36) million deaths were avoidable in 2017 by eliminating fossil-fuel combustion (27.3% of the total PM$_{2.5}$ burden), with coal contributing to over half. Other dominant global sources included residential (0.74 [0.52–0.95] million deaths; 19.2%), industrial (0.45 [0.32–0.58] million deaths; 11.7%), and energy (0.39 [0.28–0.51] million deaths; 10.2%) sectors. Our results show that regions with large anthropogenic contributions generally had the highest attributable deaths, suggesting substantial health benefits from replacing traditional energy sources.

[1] Department of Energy, Environmental, and Chemical Engineering, Washington University in St. Louis, St. Louis, MO, USA. [2] Department of Physics and Atmospheric Science, Dalhousie University, Halifax, NS, Canada. [3] Spadaro Environmental Research Consultants (SERC), Philadelphia, PA, USA. [4] Institute for Health Metrics and Evaluation, University of Washington, Seattle, WA, USA. [5] Joint Global Change Research Institute, Pacific Northwest National Laboratory, College Park, MD, USA. [6] Department of Atmospheric Sciences, University of Washington, Seattle, WA, USA. [7] Atmospheric Sciences Research Center, University at Albany, Albany, NY, USA. [8] Department of Atmospheric and Oceanic Sciences, School of Physics, Peking University, Beijing, China. [9] School of Population and Public Health, University of British Columbia, Vancouver, BC, Canada. [10]Present address: Harvard John A. Paulson School of Engineering and Applied Sciences, Harvard University, Cambridge, MA, USA. ✉email: erin.mcduffie@wustl.edu

Long-term exposure to ambient (outdoor) fine particulate matter less than 2.5 μm in diameter (PM$_{2.5}$) is the largest environmental risk factor for human health, with an estimated 4.1 million attributable deaths worldwide (7.3% of the total number of global deaths) in 2019[1]. Outdoor PM$_{2.5}$ mass is primarily composed of inorganic ions, carbonaceous compounds (black and organic carbon, including secondary organic aerosol), and mineral dust. Sources include direct emissions such as forest fires and agricultural waste burning[2,3], windblown mineral dust from arid regions[4], and inefficient fuel combustion[5], as well as secondary emissions from atmospheric chemical reactions between primary gas-phase pollutant precursors. These precursors are emitted from both combustion and non-combustion processes that include residential energy use, on- and off-road vehicles, energy generation, solvent use, industrial processes, and agricultural fertilizer application[6]. Once emitted, the chemical production of PM$_{2.5}$ mass in the atmosphere is highly non-linear[7,8]. Due to its myriad of sources and complex formation chemistry, both the total mass and chemical constituents of PM$_{2.5}$ depend on local environmental conditions, dominant sources, and the magnitude of those source-specific emissions. In addition, as air pollution and atmospheric chemistry do not adhere to political boundaries[9–11], mitigation efforts require consideration of transboundary effects across multiple locations, informed by studies of PM$_{2.5}$ source contributions and the attributable disease burden across a range of sub-national to global scales.

Source contribution studies across multiple spatial scales help to inform specific mitigation strategies and prioritize limited resources for effective action[12]. A large number of previous studies have used chemical observations or dispersion-based models to quantify sources of PM$_{2.5}$ mass, but have largely focused on specific locations or short-term events[13–15]. In comparison, comprehensive assessments of the sources and impacts of PM$_{2.5}$ across large spatial scales have been relatively limited by available long-term PM$_{2.5}$ surface measurements. A recent study, for example, found that most countries between 2010 and 2016 had fewer than 10 long-term ground-based PM$_{2.5}$ monitors per million people, while 60% of all countries had no long-term monitors[16]. Therefore, to assess the global and regional PM$_{2.5}$ disease burden and its source contributions, recent studies have employed 3D chemical transport models as a means to relate changes in surface emissions to atmospheric PM$_{2.5}$ concentrations. These studies typically use adjoint models, tagged-tracer, or zero-out (brute-force) approaches to assess the influence of individual surface sources on PM$_{2.5}$ mass and attributable mortality and morbidity. These previous studies, however, have largely focused on individual cities, countries, regions[11,17–24], or source sectors[3,25–35], often with relatively coarse spatial resolution and emissions that may not reflect current conditions. In contrast, global-scale studies that account for transboundary effects using both consistent methodologies and sectoral definitions across all world regions help to place air pollution in a global context and allow for comparability of the disease burden and its source contributions across multiple locations. Relatively few of these previous global studies, however, have provided an assessment of the contributions from more than one source sector or aggregate fuel category in recent years[36–40], thereby limiting their ability to inform or prioritize specific air quality management policies under current global conditions.

In today's rapidly changing society, the accuracy and policy relevance of such global studies is contingent on (1) the availability of contemporary and detailed emission inventories, (2) scientifically rigorous chemical transport models, (3) global fine resolution PM$_{2.5}$ exposure estimates, and (4) disease-specific concentration-response functions (CRFs) derived from contemporary air pollution epidemiologic studies. First, emission datasets that capture recent trends are particularly important in highly polluted regions, such as China, India, and Africa, that have experienced large and rapid changes in PM$_{2.5}$ precursor emissions in the last decade[6,41,42]. Disaggregation of these emissions across multiple sectors, fuel types, and regions also increases their policy relevance, as detailed source contribution studies can quantify the health benefits from specific and achievable strategies such as transitions away from coal use for energy generation or solid biofuel for residential cooking and heating. Second, to accurately reflect current PM$_{2.5}$ chemical production regimes under various emission scenarios, 3D atmospheric-chemical transport models require state-of-the-science chemical and physical mechanisms, evaluated against surface observations of PM$_{2.5}$ mass and composition. Third, to capture and compare national and sub-national impacts across all world regions, these studies additionally require high-resolution PM$_{2.5}$ exposure estimates, such as those that utilize recent advances in satellite retrievals, chemical transport models, and ground-based monitoring[43]. Lastly, integration of these source simulations and exposure estimates with updated disease-specific CRFs can motivate policy action by refining previous PM$_{2.5}$ disease burden estimates[37,38,44,45], incorporating spatial variation in the underlying health status and cause of death composition, and by comparably quantifying the dominant sources of this burden across global, national, and sub-national scales.

In this study, we integrate the emissions, modeling, PM$_{2.5}$ exposure, and CRF components described above to provide a globally comprehensive and contemporary source categorization of PM$_{2.5}$ mass and the attributable disease burden. In this work, we identify residential energy use, industrial processes, and energy generation as dominant sectors contributing to global PM$_{2.5}$ exposure and its attributable mortality. We also find that eliminating fossil fuel combustion emissions would substantially reduce (>25%) the global disease burden attributable to annual PM$_{2.5}$ exposure, with over half of this contribution from the combustion of coal. While the relative contributions from individual sectors and fuels vary across national and sub-national scales, the comprehensive nature of this work provides detailed source information relevant to developing PM$_{2.5}$ mitigation strategies and predicts a large potential health benefit from replacing traditional energy sources.

## Results

In this work, we couple emission sensitivity simulations using the GEOS-Chem 3D global chemical transport model with newly available high-resolution (1 km × 1 km) satellite-derived PM$_{2.5}$ exposure estimates[43], national-level baseline burden data, and updated CRFs from the 2019 Global Burden of Disease (GBD)[1]. We use these data and methods to quantify the relative contributions from 24 individual emission sectors and fuel categories to annual population-weighted mean (PWM) PM$_{2.5}$ mass concentrations and the attributable disease burden across 21 world regions, 204 countries (defined in Supplementary Table 1), and 200 sub-national areas.

**PM$_{2.5}$ exposure and attributable disease burden.** In 2017, the global PWM PM$_{2.5}$ mass concentration was 41.7 μg m$^{-3}$, with 91% of the world's population experiencing annual average concentrations higher than the World Health Organization (WHO) annual average guideline of 10 μg m$^{-3}$. As shown in Fig. 1a and b, exposures were highest in countries throughout Asia, the Middle East, and Africa. To maintain consistency with the GBD[1], we use the same gridded (~10 × 10 km) outdoor PM$_{2.5}$ concentration estimates[43,47,48], further downscaled to a spatial resolution of 0.01° × 0.01° (~1 × 1 km) using a newly and publicly

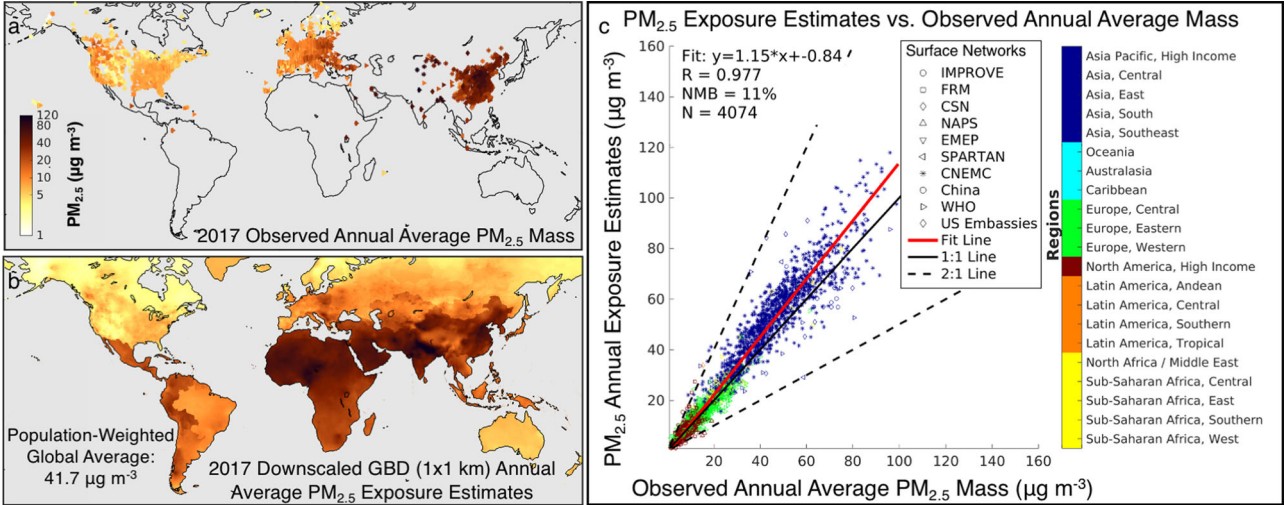

**Fig. 1 Evaluation of PM$_{2.5}$ exposure estimates relative to surface observations. a** Annual average observations of total PM$_{2.5}$ mass in the year 2017; symbol shapes correspond to monitor network. **b** Annual PM$_{2.5}$ exposure estimates for 2017, downscaled to 0.01° × 0.01° resolution. **c** Correlation between the 2017 exposure estimates and observed annual average concentrations, colored by Global Burden of Disease (GBD) region (Supplementary Table 1); symbol shape corresponds to the observation network; correlation slope, intercept, coefficient, normalized mean bias (NMB), and number of observation points are provided. (NMB = 100*Σ (exposure estimate−observations)/Σ observations).

available high-resolution satellite-derived product[43] (Methods). Figure 1c compares the resulting downscaled (~1 × 1 km) PM$_{2.5}$ concentrations to all readily available 2017 annual surface observations ($N = 4074$) of total PM$_{2.5}$ mass. Though annual surface observations are largely limited to regions in North America, Europe, and Asia, the downscaled estimates in Fig. 1c are consistent with co-located annual average observations, with a correlation coefficient ($r$) of 0.977 and a normalized mean bias of +11% or 4.6 µg m$^{−3}$.

The global ambient PM$_{2.5}$ disease burden was estimated by integrating national-level annual PWM PM$_{2.5}$ concentrations with CRFs[49] and national baseline data consistent with the 2019 GBD[1] (GBD2019 CRF). These updated CRFs better reflect the uncertainty of health effects at high PM$_{2.5}$ concentrations. Globally, we estimate that 3.83 million deaths (95% Confidence Interval: 2.72–4.97 million) were attributable to annual ambient PM$_{2.5}$ exposure in the year 2017 (Fig. 2: top left panel). Attributable deaths were primarily from ischemic heart disease (IHD) and Stroke (63%; Fig. 2: top left, right pie chart), followed by chronic obstructive pulmonary disease (COPD), lung cancer (LC), lower respiratory infections (LRI), and type II diabetes (DM). In addition, there were a total of 2.07 (95% CI: 0.02–5.02) million attributable incidences of neonatal disorders (low birth weight (LBW) and pre-term births (PTB)) worldwide (Supplementary Data 1). National-level results for 204 countries are provided in the center map of Fig. 2 (and Supplementary Data 1). The largest numbers of attributable deaths occurred in China (~1.4 [95% CI: 1.05–1.70] million) and India (0.87 [95% CI: 0.68–1.04] million), together accounting for 58% of the global total ambient PM$_{2.5}$ mortality burden. The larger burden in China, despite a lower national PM$_{2.5}$ exposure level reflects differences in population age distribution and the relative baselines associated with each disease in each country (Supplementary Fig. 1). Figure 2 also shows a large PM$_{2.5}$ disease burden in countries such as the U.S. where country-level PWM PM$_{2.5}$ exposure levels are below the WHO guideline, highlighting the risks associated with PM$_{2.5}$ exposures below 10 µg m$^{−3}$ but above the GBD counterfactual[50] (Supplementary Fig. 2; Methods). Supplementary Data 1 provides all national exposure and disease burden estimates, as well as fractional disease contributions.

As an additional sensitivity test, exposure and burden estimates for the year 2019 were additionally calculated with publicly available 2019 exposure estimates and national-level baseline burden data (Supplementary Text 1). No change was found in the global PWM PM$_{2.5}$ concentration (Supplementary Data 3), however due to changes in population characteristics (i.e., size and age decomposition in a particular country), the attributable deaths increased from 3.8 (95% CI: 2.72–4.97) million to 4.1 (95% CI: 2.9–5.3) million in 2019 (consistent with GBD2019[1]) (Supplementary Text 1; Supplementary Data 3). Disease burden estimates were also calculated using CRFs from an updated version of the Global Exposure Mortality Model (GEMM)[44] (Supplementary Text 2; Supplementary Fig. 2). While the fractional disease contributions predicted by the updated GEMM were similar to those from the GBD2019 CRFs (Supplementary Fig. 3), the absolute number of attributable deaths in each country/region were nearly always larger when the GEMM was applied.

**Global and national sector and fuel-type contributions.** Figure 2 also provides the relative (fractional) contributions of emission sectors and fuel types to annual PM$_{2.5}$ exposure levels and the attributable disease burden. As described in the Methods, fractional contributions are quantified using a recently updated version of the 3D GEOS-Chem chemical transport model[46] (Supplementary Text 3), evaluated against available surface observations (Supplementary Text 4; Supplementary Figs. 4, 5) in a series of 24 sensitivity simulations (Supplementary Table 2) with a newly released global anthropogenic emissions dataset (CEDS$_{GBD-MAPS}$[6]) that includes sector- and fuel-specific emissions for the year 2017 (Supplementary Text 5; Supplementary Fig. 6).

Results in Fig. 2 (and Supplementary Data 1) show that on the global scale, roughly 40% of the PM$_{2.5}$ disease burden was attributable to residential (19.2%; 0.74 [95% CI: 0.52–0.95] million deaths), industrial (11.7%; 0.45 [0.32–0.58] million deaths), and energy (10.2%; 0.39 [0.28–0.51] million deaths) sector emissions, which are typically associated with fuel combustion[6]. To investigate combustion contributions across all sectors, the middle pie chart in Fig. 2 (and Supplementary Data 2) illustrates the potential health benefits from eliminating specific

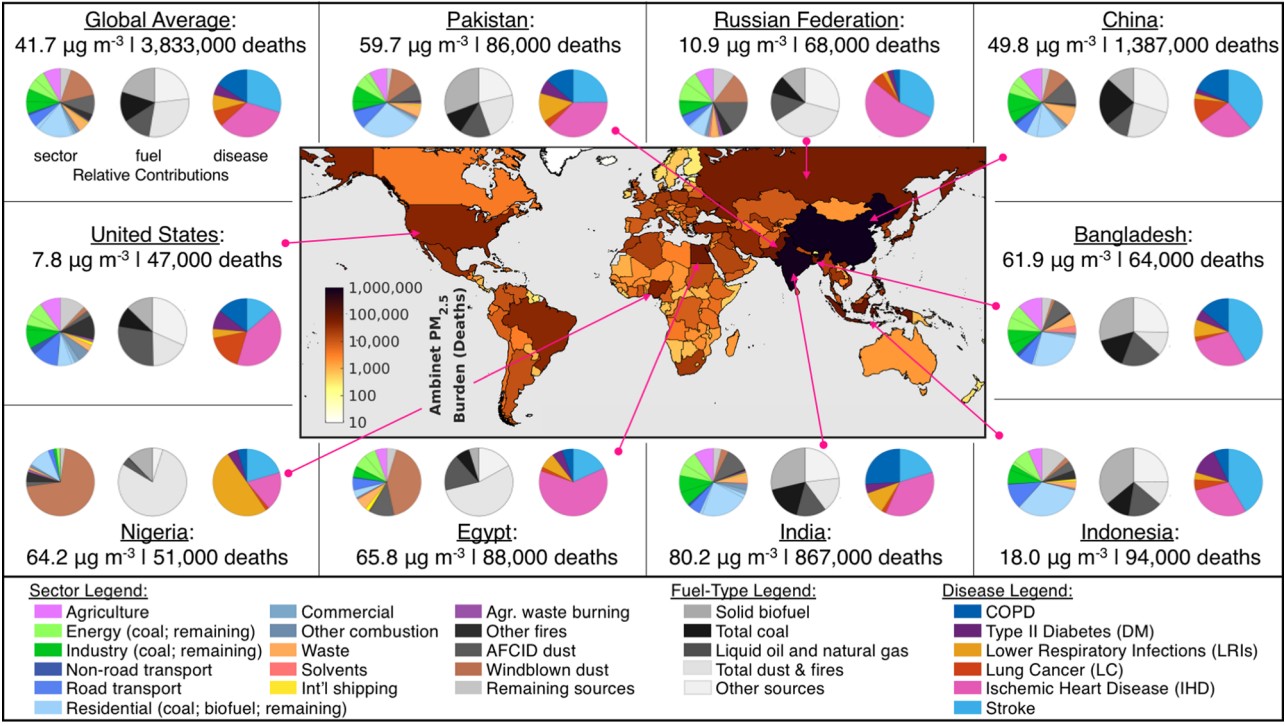

**Fig. 2 Absolute ambient PM$_{2.5}$ burden and fractional sector, fuel, and disease contributions for the global average and top nine countries.** Map: National-level outdoor PM$_{2.5}$ disease burden in 2017 (from the 2019 Global Burden of Disease concentration-response relationships). Panels: Annual average population-weighted PM$_{2.5}$ exposure levels and attributable mortality (rounded to the nearest 1000). (Left pie charts) fractional sectoral source contributions. 'Other fires' include deforestation, boreal forest, peat, savannah, and temperate forest fires. 'Remaining sources' include volcanic SO$_2$, lightning NO$_x$, biogenic soil NO, aircraft emissions, and oceanic and biogenic sources (Supplementary Table 2). Energy and industry sectors also include separate contributions from coal use (first wedge, counterclockwise). The residential sector separates the contributions from coal (first wedge) and solid biofuel (second wedge). (middle pie charts) fuel-type contributions. The 'total dust & fires' category is the sum of windblown and AFCID (anthropogenic fugitive, combustion, and industrial) dust, agricultural waste burning, and other fires. Other sources are primarily from non-combustion or uncategorized combustion sources (agriculture, solvents, biogenic SOA, waste incineration, etc.). (Right pie charts) Relative disease contributions (not including pre-term birth and low birth weight). Supplementary Data 1 and 2 provide all data in this figure, including the number of neonatal incidences.

types of combustible fuels. For example, Fig. 2 shows that nearly 1.05 (95% CI: 0.74–1.36) million or 27.3% of total PM$_{2.5}$ attributable deaths could be avoided by eliminating emissions from fossil-fuel combustion (coal = 14.1%, O&NG = 13.2%), with an additional 20% or nearly 0.77 (95% CI: 0.54–0.99) million deaths avoidable by eliminating solid biofuel combustion, primarily used for residential heating and cooking. The remaining sources in the middle pie charts largely correspond to non-combustion and natural sources, such as windblown dust, which was the second single largest sectoral source of PM$_{2.5}$ exposure at the global scale (16.1%) (Fig. 2). This source was estimated to lead to 0.62 (95% CI: 0.44–0.80) million attributable deaths worldwide under the assumption of equal toxicity of all PM$_{2.5}$ sources and components (Discussion). Other PM$_{2.5}$ sources such as on-road transportation, non-combustion agriculture emissions, and anthropogenic dust each had relatively smaller global contributions ranging between 6.0 and 9.3% (0.23 [95% CI: 0.16–0.30] to 0.36 [0.25–0.46] million deaths). Additional global source sectors, including solvents, shipping, and natural sources such as fires, biogenic, and soil emissions each contributed to less than 5.2% of the annual global PWM PM$_{2.5}$ mass. Supplementary Data 1 and 2 provide a complete data set of the global fractional sector and fuel contributions.

While global contributions provide a snapshot of globally important sectors and fuel-types, regional and country-level contributions provide information more relevant to local sources of ambient PM$_{2.5}$ mass. Therefore, Fig. 2 additionally shows the relative contributions for nine countries with the largest number

of attributable deaths associated with long-term ambient PM$_{2.5}$ exposure (from the GBD2019 CRFs). These top countries differ from those with the highest PWM PM$_{2.5}$ concentrations (Supplementary Data 1), highlighting the importance of demographic factors and disease-specific baseline estimates in calculating the total burden of disease. The majority of attributable deaths in these countries were from Stroke and IHD, except for Nigeria, where childhood LRIs were the largest cause of mortality attributable to ambient PM$_{2.5}$ exposure. Sectoral pie charts in Fig. 2 show that source contributions varied between countries, with residential contributions ranging from 4.0% in Egypt to 33.1% in Indonesia, while the sum of energy and industry emissions ranged from 3.2% in Nigeria to 27.3% in India. Windblown dust was the most variable sector within these countries, ranging from 1.5% in Bangladesh to 70.6% in Nigeria. Of the three anthropogenic fuel categories (coal, oil & natural gas, and solid biofuel), coal was the largest source of PM$_{2.5}$ attributable mortality in China (22.7%; 315,000 [95% CI: 239,000–385,000] deaths), O&NG was the largest contributor in Egypt, Russia, and the United States (13.7–27.9%; 9000 [4000–16,000] to 13,000 [4500–24,000] deaths), and solid biofuel combustion was largest in the remaining five countries (12.3–36.0%; 6000 [4500–8000] to 250,000 [196,500–300,000] deaths). Results further show that use of these fuels in China and India alone each contribute to roughly 10% of the global ambient PM$_{2.5}$ disease burden (Supplementary Data 2).

For a more holistic world view, bar charts in Fig. 3 show the relative sector and fuel contributions for all 21 world regions and

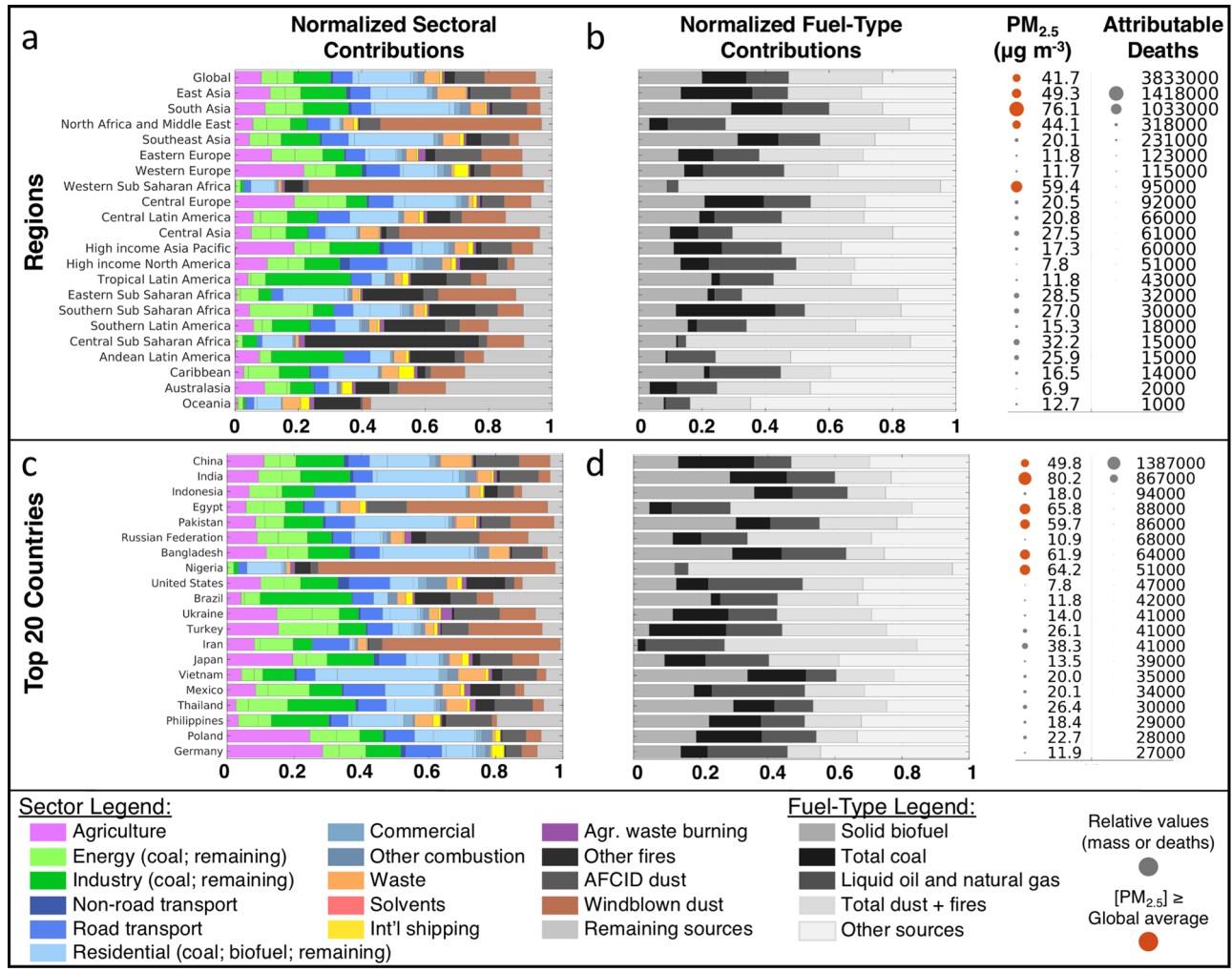

**Fig. 3 Relative (fractional) source and fuel contributions to annual population-weighted mean PM$_{2.5}$ mass and attributable deaths in 2017.**
**a**, **c** Normalized sectoral source contributions for 21 world regions and the global average (**a**) and top 20 countries (**c**). Sorted by decreasing number of ambient PM$_{2.5}$-attributable deaths (rounded to the nearest 1000). **b**, **d** Normalized contributions from the combustion of three fuel categories and remaining PM$_{2.5}$ sources. To the right of **b** and **d**, annual population-weighted mean PM$_{2.5}$ concentrations and associated attributable deaths are provided for each region/country. Relative amounts are illustrated by relative dot sizes. Concentrations above or equal to the global average are colored red.

the top 20 countries with the largest number of PM$_{2.5}$-attributable deaths. The relative contributions of PM$_{2.5}$-disease pairs for these same regions and countries are shown in Supplementary Fig. 3. The color scheme to the right of panels b and d in Fig. 3 shows that four of the 21 regions and six of the top 20 countries each had PWM PM$_{2.5}$ concentrations higher than the global average. Similar to the pie charts in Fig. 2, Fig. 3 panels a and c show that residential energy use was the largest contributing sector in South, East, and Southeast Asia, largely driven by trends in India, Indonesia, Pakistan, Bangladesh, and Vietnam. Other notable features include the dominant contribution from windblown dust throughout North Africa, the Middle East, Central Asia, and Western Sub-Saharan Africa, as well as dominant fire contributions in Southern Latin America, Central Sub-Saharan Africa, Oceania, and North America. Large agricultural contributions were found in Western and Central Europe and Pacific Asia, along with dominant contributions from industrial processes in Andean and Tropical Latin America. Comparing all world regions, Fig. 3a shows that areas with the lowest number of PM$_{2.5}$ attributable deaths generally had the smallest relative contributions from non-natural PM$_{2.5}$ sources. Similarly, Fig. 3b shows that regions with greater attributable deaths had relatively larger contributions from anthropogenic fuel combustion emissions. Exceptions include

Western and Central Sub-Saharan Africa, where combined contributions from windblown dust and fires were greatest (81.0 and 68.4%).

Figure 4a provides a map of the dominant contributing fuel type in each country to further highlight the national-level variability in relative fuel contributions. For example, Fig. 4a shows that despite a recent decline in global coal emissions[6], coal was the dominant combustible fuel type contributing to the PM$_{2.5}$ disease burden in 20 countries, including China, Eswatini, South Africa, and countries throughout Central and Eastern Europe. At the national level, South Africa and neighboring Eswatini both had the largest relative coal contributions of all countries at more than 36.5% each (~9000 [95% CI: 6000–12,500] attributable deaths in total). Countries with the lowest relative coal contributions (<0.1%) included those in other regions of Africa, as well as small island nations. O&NG combustion typically dominated in more developed countries throughout North America, Australasia, and Western Europe, as well as parts of North Africa, the Middle East, Central Asia, and Eastern Europe. Of all world regions, North America and Western Europe had the largest relative O&NG contributions at ~25% each (43,000 [95% CI: 19,500–72,500] deaths total), while the lowest was in Central Sub-Saharan Africa at 2.5% (<1000 deaths total). Third, regional

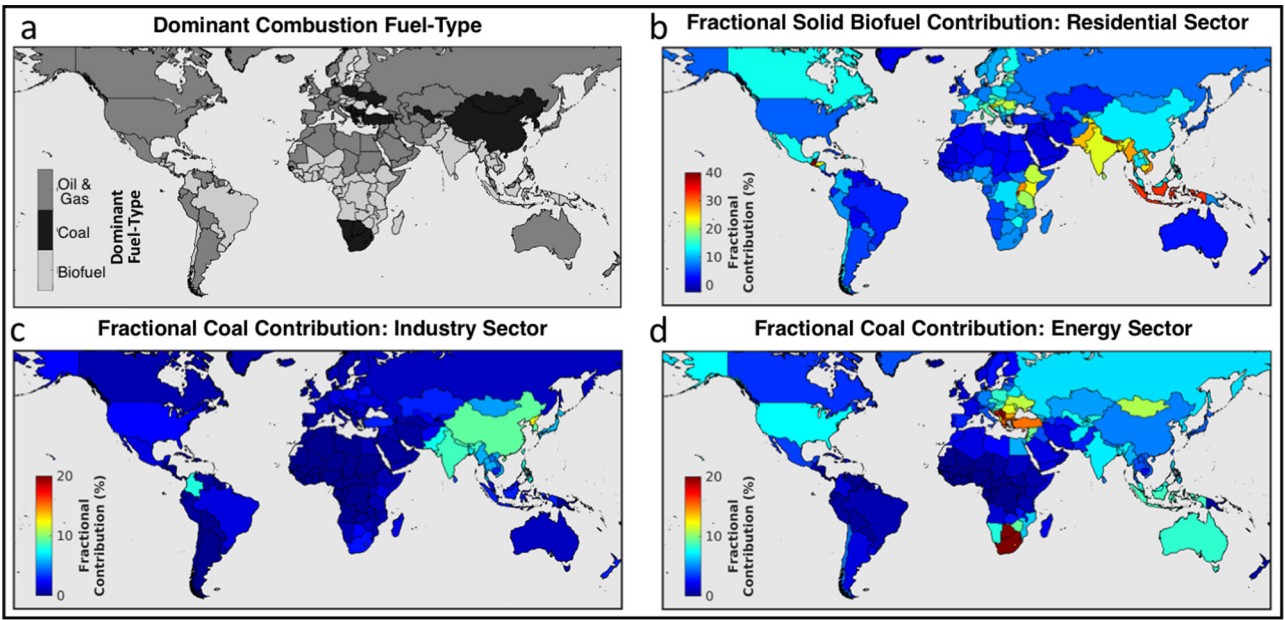

**Fig. 4 Fractional contributions from select combustion fuel types and sectors. a** The combustion fuel-type with the largest relative contribution to PM$_{2.5}$ mass and mortality in each country. **b–d** The fractional contributions from solid biofuel combustion in the residential sector (**b**), coal combustion in the industry sector (**c**), and coal combustion in the energy sector (**d**). Note the color scale change between (**b**) and (**c**, **d**).

solid biofuel contributions (largely from the residential sector) were largest in South and Southeast Asia at between 29.2 and 31.2% each (373,500 [95% CI: 279,500–465,000] deaths total). Solid biofuel was the dominant contributing combustible fuel in 76 countries including throughout Central, Eastern, and Sub-Saharan Africa, parts of Central and Western Europe, Asia, and Tropical Latin America. National-level fractional contributions ranged from 0.2% in small island nations to at least 40% in Guatemala, Nepal, and Rwanda (8500 [95% CI: 6500–11,000] total deaths).

Figure 4b–d additionally provides an assessment of three detailed emission reduction strategies that test policy-relevant scenarios of select fuel and sector combinations. These panels show the fractional contributions of PM$_{2.5}$ mass and attributable mortality avoidable by eliminating the use of (b) residential biofuel, (c) industrial sector coal combustion, and (d) coal combustion for energy generation. Figure 4a reveals that while coal is the dominant fuel type in both China and South Africa, coal from the energy sector contributes to a greater fraction of attributable deaths (20.5%) in South Africa than does the industry sector (2.7%), while the opposite is true for China (4.7% energy coal, 9.1% industry coal). Similarly, in countries throughout Central and Eastern Europe where coal is the dominant contributing fuel, the targeted reduction of coal use in the energy sector may lead to immediately larger air quality benefits than targeting coal use in the industrial sector (Fig. 4c and d). For residential biofuel use, the relative contributions are generally largest in regions where residential emissions are the dominant source sector (Fig. 3a). At the national scale, the combustion of solid biofuel for home heating and cooking contributed up to 46.1% of the total PM$_{2.5}$ mass and attributable deaths in Guatemala (Supplementary Data 1). These examples highlight the potential air quality benefits from specific and achievable reduction strategies. Detailed comparisons across countries in Fig. 4 and Supplementary Data 1 can further identify opportunities with the greatest potential health gains and identify countries who have successfully managed reductions from these select sources.

**Sub-national source contributions.** To investigate sub-national variability in PM$_{2.5}$ mass and its sources, we leverage the high-resolution downscaled exposure estimates (0.01° × 0.01°) to estimate PWM PM$_{2.5}$ mass for 200 sub-national areas. We additionally apply gridded model sensitivity simulation results (0.5° × 0.625° in North America, Europe, and East Asia, 2° × 2.5° elsewhere) to the sub-national exposure levels to estimate the relative source contributions in these same areas. Sub-national area boundaries are identified by the nearest dominant city and are defined using T3 urban extent data from the Atlas of Urban Expansion[51]. This dataset provides urban boundaries for 200 metropolitan areas that had more than 100,000 inhabitants in 2010.

PM$_{2.5}$ exposure estimates reveal a large public health benefit from reducing PM$_{2.5}$ exposure in urban areas. For example, Supplementary Data 1 shows that more than 65% of the select 200 areas experienced higher PWM PM$_{2.5}$ concentrations than their corresponding national averages. In a few extreme cases in India, for example, average PWM PM$_{2.5}$ concentrations exceeded 150 μg m$^{-3}$, nearly twice that of the national average and over 15 times larger than the WHO guideline.

Figure 5 shows that both PWM PM$_{2.5}$ mass and dominant PM$_{2.5}$ surface sources vary at the sub-national scale, highlighting the importance of developing region-specific air quality strategies. For example, while residential emissions are the largest source of average PM$_{2.5}$ exposure and attributable mortality in China and India, areas surrounding Beijing and Singrauli (Madhya Pradesh, India) have relatively larger contributions from the energy and industry sectors. Similarly, while the transportation sector was the largest PWM PM$_{2.5}$ source in the U.S., Fig. 5c illustrates regionally varying sources, with dominant contributions from forest fires in the west, windblown dust in the arid southwest, agricultural, on-road transportation, and energy throughout the midwest and east coast, and highly uncertain sources such as secondary organic aerosol (SOA) in the southeast. In Europe, the non-combustion agriculture sector is a dominant source of PWM PM$_{2.5}$ mass and mortality across large portions of the region, however pie charts in Fig. 5d also illustrate areas with relatively

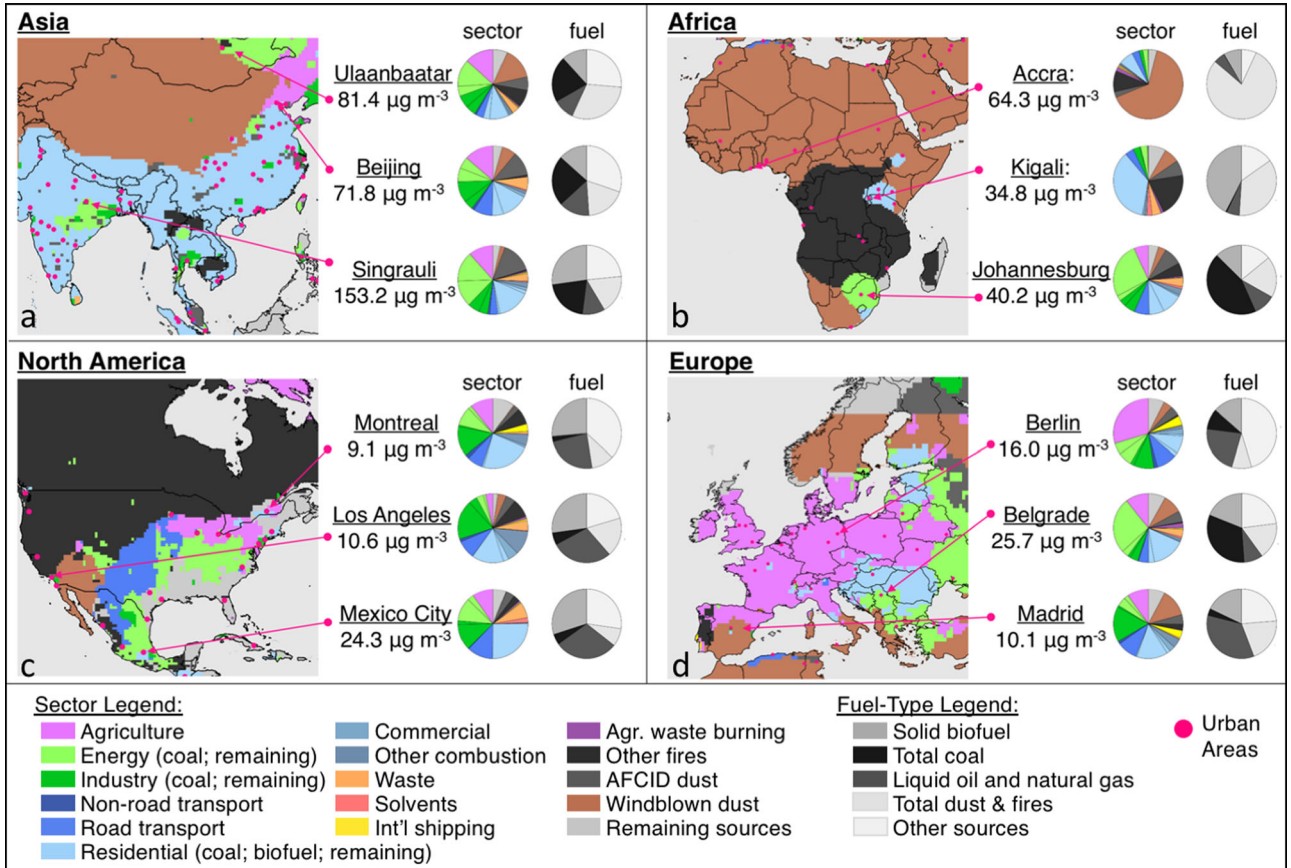

**Fig. 5 Sub-national sources of PM$_{2.5}$ mass and attributable mortality.** Results are shown for (**a**) Asia, (**b**) Africa, (**c**) North America, and (**d**) Europe. Maps illustrate the single source with the largest contribution in each model grid cell (0.5° × 0.625°). Population-weighted mean PM$_{2.5}$ concentrations (calculated from 0.01° × 0.01° PM$_{2.5}$ exposure estimates) and regional fractional source contributions are also shown for a select sub-set of sub-national regions, identified by the name of the nearest major city.

large contributions from the energy, industry, and residential sectors. For Africa, energy generation from coal combustion was the largest source of PM$_{2.5}$ attributable deaths in South Africa (Supplementary Data 1), though Fig. 5b shows that this influence was centered around Johannesburg (26.1%), while the area around Port Elizabeth was dominantly influenced by windblown dust and other non-combustion sources (Supplementary Data 1). Pie charts in all panels of Fig. 5 highlight that in all regions, a large number of sources collectively contribute to sub-national PM$_{2.5}$ mass formation, not only the largest sources illustrated in the map panels.

While annual PM$_{2.5}$ exposure estimates in Fig. 5 were derived from urban-relevant spatial scales (i.e., 0.01° × 0.01°), we note that the source contributions here are limited by the resolution of the GEOS-Chem model (above) and of the emissions dataset (0.5° × 0.5°). As a result, source contributions maps in Fig. 5 are effective at highlighting sub-national source contributions, but urban-level contributions would be improved with more spatially resolved simulations and emissions. Sub-national and urban scale PM$_{2.5}$ exposure estimates and fractional source contributions are vital for identifying reduction strategies with the greatest public health benefit, which will become increasingly important as ~65% of the world's population is projected to live in urban areas by 2050[52].

## Discussion

We provide a comprehensive and quantitative evaluation of the individual sector and fuel contributions to annual PWM PM$_{2.5}$ mass and its disease burden, relevant to the development and

prioritization of effective mitigation strategies. We find that over 1 million (27.3%) attributable deaths were avoidable by eliminating PM$_{2.5}$ mass associated with emissions from fossil-fuel combustion (total coal + O&NG). These results add to the growing evidence of the public health benefit achievable from global decarbonization strategies[53]. While global total coal contributions (14.1%) were slightly larger than those from O&NG (13.2%), the relative balance between these two fuel categories varied at the regional, country (Figs. 2 and 3), and sub-national levels (Fig. 5). As the largest number of PM$_{2.5}$ attributable deaths occurred in China and India, complete elimination of coal and O&NG combustion in these two countries could reduce the global PM$_{2.5}$ disease burden by nearly 20% (Supplementary Data 2).

Comparisons here with prior analyses are limited by differences in estimation years as well as differences in spatial resolution and input data, including chemical transport models, CRFs, and emissions, population, exposure, and burden datasets. Fractional fossil-fuel contributions to the PM$_{2.5}$ disease burden (27.3%) for the year 2017 were lower than the only previous global fractional estimate of 41% for the year 2015[36]. Observed differences are largely driven by countries that have experienced recent reductions in fossil-fuel emissions[6], such as China, the U.S., and Western European countries, including Germany and Italy. Absolute contributions in this work were also lower than recent fossil fuel attributable mortality estimates derived using different CRFs[54]. Compared to two previous national-level studies, fractional coal contributions in 2017 were also 17% smaller than a 2013 estimate for China[17], but generally consistent to

within 1% for a 2015 estimate for India[18] (Supplementary Text 6). Emission inputs suggest that $PM_{2.5}$ precursor emissions from coal combustion (e.g., $SO_2$) have decreased by up to 60% between 2013 and 2017 in China, while these same emission sources in India have increased by up to 7% between 2015 and 2017[6]. Fossil-fuel contributions in our analysis may also be lower limits as some sub-sectoral emission categories such as flaring and fossil-fuel fires were not assigned to a fuel category in the emissions dataset[6], but rather were included in the 'other sources' category in this analysis (Supplementary Text 5).

The use of solid biofuel across all sectors in 2017 contributed to an additional 767,000 (95% CI: 543,000–994,500) attributable deaths worldwide (20%), with this source in India and China again responsible for roughly 11% of the global $PM_{2.5}$ disease burden. Solid biofuel emissions in countries throughout South and Southeast Asia, as well as Central and Western Sub-Saharan Africa were largely associated with residential solid biofuel use for household heating and cooking (Fig. 5b). Large fractional contributions of this source were consistent to within 4% of the only previous global estimate[25]. Results in 2017 were also consistent to within 3% of two previous national-level estimates of fractional $PM_{2.5}$ disease burden contributions from residential heating and cooking in China in 2013[17] and in India in 2015[18] (Supplementary Text 6). While emissions from biofuel combustion have recently decreased in China, other world regions are experiencing a simultaneous increase[6], highlighting the continued importance of considering residential solid biofuel emissions for future air quality improvement strategies. Considerations of net air quality benefits will also be important in regions where a transition from residential solid biofuel use to fossil fuel energy sources may lead to immediate indoor and outdoor air quality improvements and health benefits[55], while at the same time increasing the relative fossil fuel contributions.

For major contributing global source sectors (Fig. 2), relative contributions were generally consistent with previous global studies, though differences again may arise due to real temporal changes or differences in input datasets, chemical transport models, or sectoral definitions used. Comparisons with previous national-level studies are more variable, with summaries provided in Supplementary Text 6. At the global scale, the residential energy sector was the single largest contributing source to the 2017 global $PM_{2.5}$ disease burden and had a relative contribution (~20%) similar to previous global estimates of 8–31% in 2000–2014[37–39,56]. In addition, the global 2017 contribution from the combined energy (10.2%) and industry (11.7%) sectors was consistent with two previous global combined estimates of 21 and 33% in 2010–2014[37,39]. Different reported relative contributions between the energy and industry sectors may be driven by differences in sectoral definitions. Global estimates for fractional contributions from dust and the transport sector were also generally consistent with previous global estimates. The total 2017 dust contribution (windblown and anthropogenic) of 25% was similar to two previous global studies of 18–24%[37,39]. Similarly, previous estimates for the transportation sector of between 5–12% in 2005–2015[32,37–39] encompass the 2017 value of 7.6%. In contrast, contributions from the agriculture sector (8%) were slightly lower than previous global estimates of 9–25% in 2010–2015[37,39,57], even when 2017 contributions from agricultural waste burning (+1%) were included.

Smaller global sectors include waste, fires, solvent use, and international shipping, which each contributed to <5% of the 2017 global $PM_{2.5}$ disease burden. These sources however may be important to consider for national and sub-national control strategies (Fig. 5; Supplementary Text 6). The number of global attributable deaths from the waste sector (184,000; 95% CI: 130,500–238,500 deaths) in 2017 was 30% lower than the only

previous estimate of domestic waste burning[29]. Global contributions from solvent use have not been previously reported. For international shipping, global mortality estimates (27,000; 95% CI: 19,000–35,000 deaths) fell within the range of a previous 2002 estimate[33], but were 75–95% lower than a more recent study, largely due to differences in the CRFs[58]. In contrast to anthropogenic sources, annual open fire contributions are expected to vary strongly with annual fire activity[3], and may increase in regions where the number and severity of wildfires is projected to increase[59]. The global 2017 total fire contribution was 4.1%, consistent with two previous estimates of ~5% in 2010 and 2014[37,39]. Figure 5, however, shows that fires were the single largest contributor to the $PM_{2.5}$ disease burden in select regions throughout North America, Southeast Asia, and Africa. These relative contributions are generally consistent with a previous global-scale estimate[3], though mixed agreement with previous national-level estimates likely highlights the interannual variability of this source. For example, 2017 contributions in India and China (~1%) were lower than previous estimates of 1–8% for 2013–2015[17,18,28,37], contributions in the U.S. (~13%) were more than double a previous 2010 estimate[37], and contributions in Canada (18.9%) were consistent with a previous 2013 study[11]. All remaining $PM_{2.5}$ sources (Supplementary Table 2) contributed to 5.2% or less at the global scale.

Similar to previous studies, fractional and absolute source contributions to PWM $PM_{2.5}$ mass and the attributable disease burden are subject to uncertainties in the emissions dataset, $PM_{2.5}$ exposure estimates, 3D chemical-transport model, national-level baseline mortality estimates, and the disease-specific GBD2019 CRFs. Following methods from previous similar studies[36,37,60], the 95% CI of the 2017 $PM_{2.5}$ disease burden is derived from uncertainties in the GBD2019 CRFs, resulting in a range of 2.72 million–4.97 million global attributable deaths. An additional sensitivity study is presented in Supplementary Text 7 to test the impact of uncertainties associated with the baseline mortality data, which for the majority of world regions results in smaller uncertainty bounds than those associated with CRF uncertainties (Supplementary Fig. 7). As described in the Methods, the GEOS-Chem model is evaluated against available surface observations and uncertainties in the emissions dataset are discussed elsewhere[6]. In addition, sub-national fractional source contributions (Fig. 5) are limited to the resolution of the model and emissions, while the urban exposure estimates are further subject to greater uncertainties in the satellite-derived products for small spatial scales[43,47]. Future developments of global high-resolution simulations, as well as increasing the accuracy and precision of satellite-derived $PM_{2.5}$ estimates will serve to reduce these uncertainties in $PM_{2.5}$ mass and source contributions at both the national and sub-national scales.

In addition to uncertainties in the general methodology, this work also assumes equitoxicity of aerosol mass and its sources, including from windblown mineral dust[61]. This assumption is necessary for use with the GBD2019 and GEMM CRFs and is consistent with US EPA[62] and WHO[63] assessments. This assumption may under- or over-estimate the relative $PM_{2.5}$ burden contributions from select sectors provided they contribute to more or less toxic components of total $PM_{2.5}$ mass. We additionally note that by simultaneously reducing emissions across all geographic regions, this study did not explicitly investigate national contributions from long-range or regional transport[11,64]. As the implementation of mitigation policies is typically constrained to political borders, specific policies may need to consider the regional influence on local pollution levels. We lastly note that results from the sensitivity simulations (Methods) largely reflect changes in $PM_{2.5}$ mass associated with the complete elimination of each individual emission source. Therefore, the

same relative contributions may not be expected from studies that test more moderate reduction strategies or simultaneous reductions of multiple sources (Supplementary Text 8).

The comprehensive nature of our analysis provides detailed source information to inform $PM_{2.5}$ mitigation strategies and provides potential health benefit estimates to further motivate action. Results show that residential, energy, industry, and total dust sources are among the largest contributing sectors to the global $PM_{2.5}$ disease burden, while the relative contributions from individual sources and fuels vary at the national and sub-national levels. Roughly 1 million deaths could be avoided by the global elimination of fossil-fuel combustion, with 20% of this burden associated with fossil-fuel use in China and India alone (Fig. 2). Despite recent global reductions in air pollutant emissions from coal, this fuel was still the dominant contributing combustible fuel type to the $PM_{2.5}$ disease burden in 20 countries, including China and countries throughout Southern Sub-Saharan Africa and Central Europe (Fig. 4). The use of solid biofuel was a primary source of emissions from the residential sector and was the dominant contributing combustible fuel in 78 countries, especially throughout the tropics (Fig. 4). While natural sources of $PM_{2.5}$ mass dominantly contributed in more arid regions (Fig. 3), countries with the greatest $PM_{2.5}$ disease burden generally had the largest relative contributions from anthropogenic sources, demonstrating a clear path towards attaining global air quality improvements.

## Methods

This study integrates newly available high-resolution satellite-derived $PM_{2.5}$ exposure estimates, CRFs from the 2019 GBD, and fractional source contribution results from 24 emission sensitivity simulations to provide the most comprehensive global source contribution results to-date. This work also provides global estimates of $PM_{2.5}$-attributable deaths from the use of coal, O&NG, and solid biofuel. The following sections describe the details of the high-resolution $PM_{2.5}$ exposure estimates, attributable disease burden calculations, set-up and evaluation of the chemical transport model, sector- and fuel-specific emissions dataset, and fractional simulated source contribution calculations. A schematic of this overall process is provided in Supplementary Text 9 and Supplementary Fig. 8.

**High-resolution $PM_{2.5}$ exposure estimates.** To maintain consistency with the GBD project, while also improving the accuracy of the population-exposure estimates, we downscale the 2019 GBD exposure estimates[1,47,48] to a 0.01° × 0.01° (~1 km × 1 km) grid using a newly available high-resolution $PM_{2.5}$ dataset from Hammer et al.[43]. Supplementary Text 10 (Supplementary Fig. 9) describes this process of spatial downscaling by incorporating the spatial information from the Hammer et al.[43] product. This downscaling process is independent of the modeled fractional source contribution results and maintains the average $PM_{2.5}$ mass concentration (area average only) from the original GBD product. The sensitivity of the $PM_{2.5}$ exposure estimates to the downscaling process are evaluated in Supplementary Text 10 and Supplementary Fig. 10. Exposure estimates for the year 2019 were derived using these same methods with both GBD and Hammer et al.[43] data for the year 2019.

**National-level $PM_{2.5}$ disease burden.** The total disease burden from six mortality endpoints and two neonatal disorders associated with exposure to annual average outdoor $PM_{2.5}$ mass (from the downscaled GBD-product) was calculated following a similar methodology as the 2019 GBD project[1]. First, Eq. (1) was used to derive cause-specific population attributable fractions (PAFs) for each endpoint using national-level $PM_{2.5}$ concentrations (population-weighted) from the downscaled GBD exposure estimates and recently updated relative risk curves (RR*), derived using a Meta Regression-Bayesian, Regularized, Trimmed (MR-BRT) spline from the 2019 GBD (Supplementary Fig. 2)[1]. MR-BRT curves use splines with Bayesian priors, which avoids using relative risk estimates for active smoking, previously necessary to avoid over-estimation of risks at high exposure levels. These meta-regressions were applied to the latest observational cohort and case-control studies of mortality or disease incidence from outdoor $PM_{2.5}$ pollution cohort and case-control studies; cohort, case-control, and randomized-controlled trials of household use of solid fuel for cooking; as well as cohort and case-control studies of secondhand smoke.

$$PAF_{age, disease, country} = 1 - \frac{1}{RR^*_{age, disease, [PWM\ PM_{2.5}]_{country}}} \quad (1)$$

Consistent with the 2019 GBD, the resulting RR* (or CRF) values from the MR-BRT splines in Eq. (1) are gender-independent and describe the excess risk of

non-accidental mortality from adult (25 years and older) IHD, Stroke, COPD, LC, DM, and childhood and adult (under 5 years and 25 years and older) acute LRIs. Consistent with the 2019 GBD, RRs* for each disease in Eq. (1) are also a function of annual PWM $PM_{2.5}$ mass exposure in each country and the difference between this exposure level and the Theoretical Minimum Risk Exposure Level (TMREL). The TMREL in this work, as in the GBD2019, is assumed to have a uniform distribution ranging between 2.4 and 5.9 μg m$^{-3}$. Thus, RR* = RR(age, PWM $PM_{2.5}$)/RR(age, TMREL). RR* values were also stratified by quinquennial age group (25–29, …, 95+), with age-specific RR* values applied to IHD and stroke outcomes. In contrast, age-independent RR*s were applied to the other health outcomes (age group 25 and over for COPD, LC, and type II DM, and the combined age groups under 5 and 25 and over for LRI). Supplementary Fig. 2 provides an illustration of select RR* (or CRF) values for these diseases as a function of annual $PM_{2.5}$ mass exposure, as well as the CRFs for two neonatal disorders, which include the number of preterm (PTB; gestational age less than 37 weeks) and low birth weight (LBW; below 2.5 kg) incidences. The 95% CI for the CRF values was determined from the distribution of 2000 randomly selected values of the TMREL.

As shown in Eq. (2), the PAFs for each age group, disease, and country were then multiplied by the age- and country-specific baseline mortality data for each disease and summed over all relevant age groups (m) and diseases (n) to obtain the total national-level $PM_{2.5}$ burden associated with exposure to both outdoor and household (indoor) $PM_{2.5}$ mass. National cause- and age-specific baseline mortality data for the years 2017 and 2019 were extracted from the GHDx database[65]. The national-level baselines for PTB and LBW were calculated from the 2019 GBD statistics of the number of annual live births and the percentage of LBW and PTB cases at the national and regional levels[66,67].

$$PM_{2.5}\ Attributable\ Mortality_{country} = \sum_{disease}^{n} \sum_{age}^{m} PAF_{age, disease, country}$$
$$\times Baseline\ Mortality_{age, disease, country} \quad (2)$$

Finally, to separate the contributions from outdoor and indoor household co-exposure, the national-level total $PM_{2.5}$ attributable mortality values from Eq. (2) were scaled using Eq. (3) to account for the risk of co-exposure to household air pollution included in the CRFs. Country-specific adjustment factors were derived from a comparison of national-level burdens in Eq. (2) to those derived for outdoor exposure only in the 2019 GBD study. As a result of these adjustments, the $PM_{2.5}$ attributable mortality and source contribution results presented in this analysis reflect contributions from indoor sources of air pollution (e.g., biomass combustion for residential heating and cooking) to the extent that they impact ambient $PM_{2.5}$ concentrations.

$$Outdoor\ PM_{2.5}\ Attributable\ Mortality_{country} = PM_{2.5}\ Attributable\ Mortality_{country}$$
$$\times Adjustment\ Factor_{country}$$
$$(3)$$

The overall approach described here generally follows that of the 2019 GBD, but deviates in part by calculating national-level PAFs rather than PAFs for each grid cell, and by using publicly available national baseline data from the IHME[65], rather than both national and sub-national baseline estimates. We find that the aggregate-country level method used in this work is consistent to within 5% of the grid-cell methodology used in the GBD.

As discussed in Supplementary Text 2, we also calculate the PAFs for each $PM_{2.5}$-disease pair (plus two neonatal disorders) using an updated version of the GEMM[44]. To aid in the comparison with GBD2019 CRF estimates, the original GEMM was updated to include CR curves for Type-II Diabetes, PTBs, and LBWs as well as newly available observational data (described in Supplementary Text 2). For the neonatal outcomes, only the number of PTB and LBW cases were estimated, whereas the 2019 GBD estimated neonatal death mediated by the impact of $PM_{2.5}$ on birthweight and short gestation. As the GEMM is exclusively developed from studies of outdoor $PM_{2.5}$ exposure, total outdoor $PM_{2.5}$ attributable deaths in Supplementary Data 1 and 2 were taken directly from Eq. (2) and did not require scaling factors to remove the risk associated with indoor $PM_{2.5}$ exposure.

**Simulated fractional sector and fuel-type contributions.** Fractional sector and fuel-specific contributions were derived from a series of emission sensitivity simulations, using the 3D GEOS-Chem chemical transport model[46]. As described in Supplementary Text 3, we used the GEOS-Chem v12.1.0 source code, updated to account for scientific updates to physical deposition, reactive nitrogen chemistry, and surface emissions (https://github.com/emcduffie/GC_v12.1.0_EEM). Model simulations were run from December 2016 to January 2018 to allow for one month of spin-up. The model was run globally at a resolution of 2° × 2.5° and was supplemented with three nested simulations with resolutions of 0.5° × 0.625° over North America, Europe, and Asia.

Gridded emission datasets are the backbone of any modeling source contribution study. In this work, we leverage a newly developed emissions dataset developed from the Community Emissions Data System that has been updated for the GBD-MAPS project (https://sites.wustl.edu/acag/datasets/gbd-maps/)[6]. This dataset provides global gridded (0.5° × 0.5°) emissions of key $PM_{2.5}$ components (black and organic carbon) and gas-phase precursors from 11 individual

anthropogenic source sectors and multiple fuel types for the year 2017. Supplementary Fig. 6 illustrates these global emissions as a function of source sector and chemical compound. Additional emission inputs used for model sensitivity simulations largely include those from fires (forest fires and agricultural waste burning)[68], biogenic sources, and anthropogenic[69] and windblown dust. Supplementary Text 5 provides further emission details.

In source sensitivity simulations, it remains vital to evaluate the model's ability to predict total $PM_{2.5}$ concentrations as well as regional chemical production regimes. Comparisons to total $PM_{2.5}$ mass provide confidence in the model's ability to accurately simulate total mass production. Additional comparisons to $PM_{2.5}$ chemical components imply accuracy in the model's ability to capture $PM_{2.5}$ formation chemistry and provide confidence in the model's ability to accurately predict chemical changes in response to specific emission scenarios. In this work, we evaluated the base GEOS-Chem simulation (including all emission sources) against all available long-term surface observations of both $PM_{2.5}$ mass and its chemical composition. As described in Supplementary Text 4, the observational dataset was compiled from more than 10 long-term observation networks and over 4000 individual sites (Supplementary Figs. 4 and 5a).

The comparisons in Supplementary Fig. 5 indicate that individual components in the base GEOS-Chem simulation agree to within $-0.3$ to $0.6\,\mu g\,m^{-3}$ of the observed annual average concentrations for all $PM_{2.5}$ chemical components. These observations ($N < 230$) were largely limited to North America, Europe, and China, however, Supplementary Fig. 5 also demonstrates the large improvement in the long-standing bias of aerosol nitrate in our updated version of the GEOS-Chem model[70] relative to the default version. Supplementary Fig. 5 also demonstrates relative improvements in the updated model in concentrations of sulfate, ammonium, and dust. In terms of total $PM_{2.5}$ mass, Supplementary Fig. 10a shows that the base model predictions were consistent with the 2017 observations (NMB of $+5\%$, and correlation ($r$) of 0.89).

For the 24 individual emission sensitivity simulation sets (1 global +3 nested per set), we employed a zeroing out (brute force) method[20,36-38], where fractional $PM_{2.5}$ mass contributions from each source were calculated from simulations that systematically remove individual source sectors or fuel-specific emissions. Supplementary Table 2 provides a detailed list of the 24 individual sensitivity tests. Resulting simulated spatially resolved $PM_{2.5}$ mass contributions from each source category were calculated following Eq. (4) and Eq. (5), where simulated gridded total $PM_{2.5}$ mass concentrations from each sensitivity study were first compared to the total gridded $PM_{2.5}$ mass in the base simulation (4), and then were compared to the sum of $PM_{2.5}$ mass from all $j$ simulations (5) to calculate the gridded fractional contributions.

$$\left[PM_{2.5}\right]_{source} = \left[PM_{2.5}\right]_{base\ simulation} - \left[PM_{2.5}\right]_{source\ sensitivity\ simulation} \tag{4}$$

$$\left(\%PM_{2.5}\right)_{source} = \frac{\left[PM_{2.5}\right]_{source}}{\sum_{j=1}^{24}\left[PM_{2.5}\right]_j} \tag{5}$$

Lastly, Eq. (6) was used to calculate the fractional source contributions to PWM $PM_{2.5}$ mass in 200 sub-national areas, 204 countries, 21 world regions, and for the global average. First, absolute contributions from each source were calculated from the product of the spatially resolved fractional $PM_{2.5}$ source contributions from Eq. (5) and the spatially resolved downscaled GBD exposure estimates from Fig. 1, averaged over $i$ grid boxes and weighted by the total population in a given region or country. Fractional contributions were then calculated by dividing these absolute source contributions by the total PWM $PM_{2.5}$ concentration in a given region, country, or area. In Eq. (6), variable $i$ represents individual grid boxes to distinguish the use of spatially resolved vs. spatially averaged products. Population-weighted fractional source contributions were calculated at the spatial resolution of the GEOS-Chem model. Supplementary Data 1 and 2 provide the resulting population-weighted fractional source contributions from Eq. (6).

$$\left(\overline{\%Contribution}\right)_{source} = \frac{\sum_{i=1}^{n}\left(\%PM_{2.5}\right)_{source_i} \times \left[GBD\ PM_{2.5}\right]_i \times population_i}{\sum_{i=1}^{n} population_i} \Big/ \frac{\sum_{i=1}^{n}\left[GBD\ PM_{2.5}\right]_i \times population_i}{\sum_{i=1}^{n} population_i} \tag{6}$$

Following the approach of previous studies[20,37], fractional contributions from Eq. (6) are then applied to national-level $PM_{2.5}$ exposure levels (population-weighted) and total disease burden estimates (provided in Supplementary Data 1 and 2) to calculate the absolute contributions from each source (reported in the Main Text). This method eliminates the sensitivity of the burden calculation to the order in which emission sectors are removed in model sensitivity simulations.

## Data availability
Supplementary Data files 1 and 2 provide the global, regional, national, and sub-national fractional sector and fuel contributions to $PM_{2.5}$ mass and disease burden for the year 2017. Supplementary Data files 1 and 2 also provide the total disease burden estimates determined by the GBD2019 and GEMM CRFs and the fractional disease-specific contributions to each (Supplementary Data 1 only). Supplementary Data 3 provides the 2019 $PM_{2.5}$ exposure estimates. A data visualization tool for all Supplementary Data is

available at: https://gbdmaps.med.ubc.ca/. Gridded model fractional source contribution results are available at: https://zenodo.org/record/4739100. CEDS$_{GBD-MAPS}$ emissions data are available at: https://zenodo.org/record/3754964. Input datasets required for this analysis (including high-resolution exposure estimates and GBD baseline burden data and CRFs) are available at: https://zenodo.org/record/4642700.

## Code availability
The GEOS-Chem model source code used for sensitivity simulations is available at: https://github.com/emcduffie/GC_v12.1.0_EEM and https://zenodo.org/record/4718622. The CEDS source code used to develop the global emissions dataset is available at: https://github.com/emcduffie/CEDS and https://doi.org/10.5281/zenodo.3865670. The analysis scripts used in here are available at: https://github.com/emcduffie/GBD-MAPS-Global and https://zenodo.org/record/4718618.

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

## Acknowledgements

The research described in the article was conducted under contract (Grant Agreement: #4965/19-1) to the Health Effects Institute (HEI), an organization jointly funded by the U.S. Environmental Protection Agency (EPA; Assistance Award No. R-82811201) and certain motor vehicle engine manufacturers. The contents of this article do not

necessarily reflect the views of the HEI or its sponsors, nor do they necessarily reflect the views and policies of the EPA or motor vehicle and engine manufacturers. We would like to thank Chi Li, Jun Meng, Crystal Weagle, Brian Boys, and Colin Lee for support during the development of the GEOS-Chem simulations and analysis scripts.

## Author contributions

M.B., R.V.M., and E.E.M. conceived the project. E.E.M. developed and conducted the model simulations and model analysis scripts, with computing support from L.B.. E.E.M. conducted the disease burden analysis, as directed by M.B. and J.V.S., with contributions from R.B.. M.S.H. and A.v.D. provided high-resolution satellite-derived $PM_{2.5}$ exposure estimates. J.L. and J.A.A. provided compiled observations of $PM_{2.5}$ chemical components from China. E.E.M., S.J.S., and P.O. contributed to the development of the $CEDS_{GBD-MAPS}$ dataset. V.S., L.J., G.L., and F.Y. supported the update of the GEOS-Chem model. E.E.M., M.B., and R.V.M. wrote the manuscript with contributions from all co-authors.

## Competing interests

The authors declare no competing interests.
