## [Peer Review File · Nature Communications]

Reviewer comments, first round

Reviewer #1 (Remarks to the Author):

"Source Sector and Fuel Contributions to Ambient PM_{2.5} and Attributable Mortality Across Multiple Spatial Scales" by McDuffie et al., provides multiple timely and important information, it is well written and warrants publication in Nature Communication. I have a few moderate comments which the authors may want to follow-up.

- a) I would suggest inclusion of burden of excess death among neonates in the pie-charts of Fig.2.
- b) I am surprised to see desert and windblown dust contributing to just ~5% of the total death burden in India, previous region specific studies (GBD-MAPS India) estimate much higher numbers (~30%). I was wondering if the dust scheme used here is appropriate. I doubt if there are sufficient detailed representations of interaction of dust with air pollution (eg. chemical aging), which alters the microphysics of particles relevant for their atmospheric lifetime and also dust transport.
- c) I was wondering at what size range (Aitken/accumulation) the primary organic aerosols and black carbon are emitted in the model?
- d) I strongly suggest the authors to improve the representation of 95%CI ranges in the excess death estimates by combining the uncertainties in baseline disease rates and the CRFs.

Reviewer #2 (Remarks to the Author):

This manuscript analyzed global health issues by PM_{2.5} with a broader view of source sector- and fuel-specific contributions in 2017. The manuscript also forecast the avoided deaths in certain PM_{2.5} control scenario. The topic of health effect of global PM_{2.5} is relevant to the scope of Nature Communications, but there are some statements to be clarified. I would recommend this manuscript to be accepted with minor modification.

Minor comments

1. Lines 43-44, the morality should come with confidence interval. I think the CI is needed for the mortality appeared first time throughout the text.
2. Lines 167-169, I am very curious with the higher mortality in China compared with India. I feel the CI for mortality is needed here. It is better to show the magnitude differences in population age distribution and the relative baselines associated with each disease between China and India. Also, any previous studies show the same conclusion?
3. Line 176-178, is it a better way to project 2019 emission based on 2017 to meet the burden estimates for 2019? I understand this may bring additional uncertainty but at least a discussion is needed here.
4. Lines 536-539, the validation of geographical-weighted regression between AOD and PM_{2.5} is needed here at least with a summary in SI although the cited paper discussed.

Reviewer #3 (Remarks to the Author):

The MS Source Sector and Fuel Contributions to Ambient PM_{2.5} and Attributable Mortality Across Multiple Spatial Scales is overall very well written and present a comprehensive and novel global assessment of sectoral contributions to PM_{2.5} concentrations at different spatial scales.

I only have a few comments and suggestions for minor changes before I would consider this MS suitable for publication:

L50: "responsible for 4.1 million deaths" - throughout the MS, you primarily focus on association with premature deaths/mortality, and I would suggest to nuance this statement accordingly here.

L51/52: I understand your focus is on the contribution of fossil fuels, but a couple of sentences at least here (or in methodology) on the potential contribution by organic aerosol would be useful. I am not suggesting a major edit, but identifying that composition of PM_{2.5} is still uncertain in particular in relation to the contribution of SOA. I would argue that an in depth discussion of condensable fractions and detailed composition is beyond the scope of your global assessment, however. This is linked as well to L350 where you make the point of highlighting the importance of developing region-specific AQ strategies.

L81ff: I agree with your assessment that there are not many global studies, but it is not immediately clear what the global studies provide over regional or national scale studies. The key issue of transboundary effects of e.g. SIA and in particular due to ammonium nitrates and - sulfates is clear and e.g. addressed by the UNECE CLRTAP or TFTAP. Would the key argument for global studies be that such an assessment may help to avoid pollution transfer through identifying the contribution of energy-intensive industrial production to other world regions? What I am after is a brief argument in the introduction to put the global study in perspective - not just that there are few, but that they can play a vital role in international policy design.

Fig 1. This is a key output and panel C in particular is fairly dense in that it presents a lot of data dimensions. For the scatterplot, I am wondering if the shapes of observations sites (and thus the legend) could be omitted here and presented in the SI, as a key interesting element is the regional clustering, which could be drawn out better here. Either that, or increasing the size of Panel C to a full-width figure may much improve accessibility to readers.

Fig 2. Similar comments on the colouring and shading of the pie charts - It becomes fairly difficult for a reader to fully appraise the different data content where colour/shading is very similar. There is without doubt a trade-off between density of information presented and accessibility by readers here.

L227/8: the assumption of equal toxicity is appropriate here and I would challenge that at all; I am wondering if the rather different world regions and PM_{2.5} compositions assessed, however, could yield some insights if differences in premature mortality associated with different composition 'fingerprints' indicate and more or less 'toxic' mixes? Or are confounding factors too complex to draw out such a conclusion?

Fig 3. numbers in legends are very difficult to read and could benefit from revision; in addition, the largest number of attributable deaths seems to not be associated with a sized circle in Panel B?

L394/6: What role do population density play in the occurrence of the largest number of PM_{2.5} attributable deaths in China and India? For instance, programmes in India to reduce indoor burning of solid wood or waste for cooking in deprived areas and informal settlements in Africa by introducing natural gas burners may reduce premature mortality by improving indoor air quality, but could be considered negative for outdoor air quality due to being related to fossil fuel combustion. Either here or somewhere, it could be helpful to state if indoor AQ is at all considered, or not?

SI-Fig6/7. These figures and schematics try to provide an overview of the flow of data and how they are combined, which is much appreciated. I would, however, argue that they are not as clear and accessible as they could be. The use of GBD, EO and ACTM data, alongside other data sources, is indispensable, but it comes at a cost as it is not fully clear and transparent of how these sources are combined and how e.g. inherent uncertainties or ranges of data are accounted for. I would as well suggest to include a preliminary step in particular on the GEOS-CHEM modelling, as emission data inputs seem to be missing in SI-Fig 6. While I do not have a perfect solution for a clearer flowchart either, perhaps removing the maps as examples in SI-Fig 6 and instead focusing on more descriptive text or representation of the actual data types (and their sources) which are used in each stage could be a solution?

I would like to stress again that all comments are of a minor and predominantly presentational/editorial in nature. The methodology and robustness of the analysis are sound and the degree of detail presented - while it could at times be clearer - appropriate for a study of this caliber. I have very much enjoyed reading the MS and look forward to see it published hopefully soon.

Stefan Reis

Reviewer #4 (Remarks to the Author):

The title and abstract of this manuscript accurately describe the work presented. This work uses highly spatially resolved (for a global domain) modelled data of speciated PM_{2.5} concentrations and PM_{2.5}-disease pair concentration response functions (CRFs) from the Global Burden of Disease to calculate spatially-resolved numbers of attributable premature deaths across the globe. This in itself is not novel. The novelty is the assignment of attributable premature deaths to each of a series of source sectors contributing to PM_{2.5} (via primary and secondary PM_{2.5}) and, separately, the assignment of attributable premature deaths to a series of fuel types underpinning the emissions. A further novelty is the quantification of these source and fuel assignments at multiple scales of area disaggregation, from GBD region, to individual country, to sub-national and urban spatial scales.

The manuscript is extremely well presented. The description is technically focused and unambiguously written. A lot of information is presented in each figure. A comprehensive supplementary information is provided which includes further methodological descriptions and results and discussion of sensitivity studies. Data files are also provided.

The work is of very high quality, provides a lot of interesting and insightful data and is of global interest. The discussion is supported by the methods and the data presented.

The results presented do, of course, critically depend on the quality of the contributing datasets and of the atmospheric chemistry modelling. The team work at the forefront of this research area. Many of the methodological approaches, and much of the data underpinning this study, are related to the GBD framework so that the work presented here is therefore consistent with GBD output. This applies both to the PM_{2.5} modelling and to the CRFs and underlying mortality data applied. The work builds on the team's previous work and the paper is comprehensively referenced. The derivation of sector contributions to PM_{2.5} uses the zero-out or so-called brute force modelling approach. This uses model runs in which there is complete removal of all emissions from a given source. This is widely used. However, it does mean that the sum of the changes in PM_{2.5} arising from elimination of each source individually exceeds the PM_{2.5} concentration in the base simulation. This work uses the standard approach of calculating the individual source sensitivity relative to the sum of the effects from all the sources rather than relative to the baseline PM_{2.5} concentration. A more fundamental issue in this brute-force approach is that the relative contribution assigned to each source may be different to the relative contributions derived from model simulations with more moderate emissions perturbations or when multiple sources are perturbed together. The authors of this paper recognise and highlight these issues. My view is that the approach taken here is satisfactory and that the general trends revealed in this study are likely to be fine, as long as the specific numbers aren't taken too literally as absolutely correct. And the output of this study is surely much more sensitive to uncertainty in the model input emissions than to inaccuracies introduced by the brute-force approach.

I support publication of this work essentially as it is. I make only a couple of minor observations and draw attention to one presentational error.

L41: Insert the words 'annually' to read "Nearly 1.05 million deaths annually worldwide....."

Fig. 5: Maps that highlight the largest contributor to something, in this case the sector with the highest contribution to mortality, can be misleading because of the potential very different

absolute contributions of the highest contributing source in different places. For example the highest contributing source might contribute 40% in one place but only 10% somewhere else. This does not make the maps less useful, but a caveat to this effect could be included in the discussion of the figure.

Reference #42 is the same as reference #6.

Review Responses for: Source Sector and Fuel Contributions to Ambient PM_{2.5} and Attributable Mortality Across Multiple Spatial Scales by Erin E. McDuffie et al.

We thank the Reviewers for their positive and insightful comments, which have helped improve the quality and clarity of our manuscript. We have responded to each comment below. The original comments are in black, our responses are in blue and the changes to the manuscript text are in *blue italics*. Overall, the analysis remains unchanged, though we have updated the main text and supplement accordingly to address Reviewer questions and comments. Changes were made to maintain a similar manuscript length, while providing improved clarity, context, and an enhanced discussion on uncertainty. We have also made adjustments (including to the abstract) to adhere with Nature Communications formatting requirements. Line numbers in our responses below correspond to the submitted tracked changes version of the manuscript and its supplement.

Review #1

Source Sector and Fuel Contributions to Ambient PM_{2.5} and Attributable Mortality Across Multiple Spatial Scales" by McDuffie et al., provides multiple timely and important information, it is well written and warrants publication in Nature Communication. I have a few moderate comments which the authors may want to follow-up.

We thank the Reviewer for their comments and have addressed each below. Most notably, we have added an additional sensitivity analysis for the disease burden uncertainty and have provided clarification in the main text and supplement regarding the neonatal disorders and model aerosol schemes.

a) I would suggest inclusion of burden of excess death among neonates in the pie-charts of Fig.2.

In this work, we report Low Birth Weight (LBW) and Pre-Term Birth (PTB) counts as number of incidences rather than deaths. As LBW and PTB are not themselves diseases, the attributable number of neonatal deaths are mediated through these conditions. In the 2019 GBD¹, neonatal mortality attributable to PM_{2.5} that is mediated by the impact of PM_{2.5} on birth weight and short gestation were included for the first time, and include associated neonatal mortality from diarrhoeal disease, lower respiratory infections, upper respiratory infections, otitis media, meningitis, encephalitis, neonatal preterm birth, neonatal encephalopathy due to birth asphyxia and trauma, neonatal sepsis and other infections, as well as hemolytic disease and other neonatal jaundice and other disorders¹. LBW and PTB are not considered independent and to estimate the complicated relationship between incidences of these disorders and the attributable mortality associated with the abovementioned diseases, the GBD conducted additional analyses of the impact of PM_{2.5} on shifts in the birthweight and gestational age joint distribution with non-publicly released data and in part to avoid double counting with other diseases already included in the total burden, such as LRIs. This estimation process is described in detail in the Methods Appendix to the 2019 GBD risk factor publication¹. Therefore, due to the uncertainty and complicated nature of this relationship and the objective in this study to use publicly available data sources, under the neonatal category in this analysis, we calculate the number of incidences of LBW and PTB attributable to ambient PM_{2.5} exposure, not the attributable deaths mediated by birthweight and short gestation.

As the pie charts in Fig. 2 are provided to illustrate the fractional contributions of 6 PM_{2.5}-disease pairs to the total number of PM_{2.5} attributable deaths, we have not included the number of incidences of LBW and PTB, which are in different units. To clarify this point, we have added the following sentence to the main text and have added clarification in the caption of Fig. 2 to note that the number of neonatal

incidences are not included but are provided in the supplemental Data Files. We have also included the 95% confidence interval for the total number of incidence per Reviewer #1's comment D and Reviewer #2's comment 1.

Line 130 –

In addition, there were a total of 2.07 (95% CI: 0.02-5.02) million attributable incidences of neonatal disorders (Low Birth Weight (LBW) + Pre-Term Births (PTB)) worldwide (Data File 1).

Line 649

For the neonatal outcomes, only the number of PTB and LBW cases were *estimated*, whereas the 2019 GBD estimated neonatal death mediated by the impact of PM_{2.5} on birthweight and short gestation.

Fig. 2. Ambient PM_{2.5} burden and fractional source sector, fuel, and disease contributions for the global average and top nine countries. **Map:** National-level outdoor PM_{2.5} disease burden in 2017 (from GBD2019 CRF). **Panels:** Annual average population-weighted PM_{2.5} exposure levels and attributable mortality (rounded to the nearest 1000). **(Left pie charts)** fractional sectoral source contributions. ‘Other fires’ include deforestation, boreal forest, peat, savannah, and temperate forest fires. ‘Remaining sources’ include volcanic SO₂, lightning NO_x, biogenic soil NO, aircraft emissions, and oceanic and biogenic sources (SI-Table 2). Energy and industry sectors also include separate contributions from coal use (first wedge, counterclockwise). The residential sector separates the contributions from coal (first wedge) and solid biofuel (second wedge). **(Middle pie charts)** fuel-type contributions. The ‘total dust & fires’ category is the sum of windblown and AFCID (Anthropogenic Fugitive, Combustion, and Industrial) dust, agricultural waste burning, and other fires. Other sources are primarily from non-combustion or uncategorized combustion sources (agriculture, solvents, biogenic SOA, waste incineration, etc.). **(Right pie charts)** Relative disease contributions (*not including PTB and LBW*). Data Files 1 and 2 provide all data in this figure, *including the number of neonatal incidences*.

Supplement Line 77 –

For neonatal disorders, the *incidence* associated with outdoor PM_{2.5} exposure totaled to 2.07 (95% CI: 0.02-5.02) million worldwide, which increased marginally to 2.09 (95% CI: 0.02-5.06) million in 2019.

SI-Fig. 3: Normalized disease contributions to total attributable mortality in 2017 for 21 world regions (A, C) and 20 countries (B, D) with the highest outdoor PM_{2.5} disease burden. Panels show results estimated using the GBD2019 CRFs (A, B) and the updated GEMM (C, D). Bar charts show the relative contributions of six PM_{2.5}-disease pairs to regional and national-level outdoor PM_{2.5} attributable deaths, sorted by decreasing number of deaths. *The number of LBW and PTB incidences are included in Supplemental Data File 1.* PWM PM_{2.5} concentrations and number of attributable deaths are additionally provided for each region/country. *Relative amounts are illustrated by relative dot size (except for the global total disease burden).* Red dots indicate regions/countries with PM_{2.5} exposure levels equivalent or larger than the global average.

SI File Description –

SI Data File 1. Global, regional, national, and sub-national PM_{2.5} exposure estimates and sector and disease-specific fractional contributions. Provides *downscaled PWM* national-level PM_{2.5} exposure estimates for 200 sub-national areas, 204 countries and territories, and 21 world regions. This table also provides the total *attributable deaths and the number of neonatal incidences* associated with PWM PM_{2.5} exposure levels in each country and region. *Burden results* are provided from both the GBD2019 CRFs and the GEMM. *Also includes the fractional contributions* (units of percent) of each source sector and disease (COPD, DM, LRI, LC, IHD, Stroke) to the total GBD2019 and GEMM disease burden estimates (included as separate .xlsx file).

b) I am surprised to see desert and windblown dust contributing to just ~5% of the total death burden in India, previous region specific studies (GBD-MAPS India) estimate much higher numbers (~30%). I was wondering if the dust scheme used here is appropriate. I doubt if there are sufficient detailed representations of interaction of dust with air pollution (eg. chemical aging), which alters the microphysics of particles relevant for their atmospheric lifetime and also dust transport.

As the Reviewer notes, the fractional contributions from total dust emissions in India were estimated to be ~ 38% (~29% from windblown, ~9% from anthropogenic) in 2015 in the previous GBD-MAPS India study², while total dust contributions in India for this work in 2017 were lower at ~ 15% (3.4% from windblow, 11.5% from anthropogenic dust). These differences may result for a variety of reasons, including model mechanistic differences and real interannual variability in dust emissions and removal rates through precipitation. For example, the previous GBD-MAPS India study was conducted using v10 of the GEOS-Chem model, while this work used a more recent v12 of the model source code. In both versions, mineral dust emissions were calculated during each model time step using the Dust Entrainment and Deposition (DEAD) mobilization scheme³, which accounts for variables such as soil composition, land cover type, and windspeed. Anthropogenic emissions in both models were also from a recently developed inventory⁴. Two major model updates since the v10 source code likely contribute to the reduced dust contributions in India in this work. First, an improved dust size distribution scheme⁵, has been implemented into GEOS-Chem, which significantly reduced (~70-80%) model dust concentrations over the western U.S. and improved the model agreement with surface dust observations⁵. Second, as described in SI-Text 3, this work additionally implements a new wet deposition scheme⁶, which also serves to reduce dust concentrations relative to the default v12 model source code and improves agreement with available observational sites (SI- Fig. 5). In addition to model differences, interannual variability in surface dust concentrations in South Asia⁷ may also contribute to the differences between these studies.

We have adjusted the text in the SI-Text 6 where the dust contributions in India are discussed to highlight that model mechanistic updates and interannual variability likely contribute to differences in relative dust contributions between this and the previous GBD-MAPS India study. This example provides an illustration of our note on line 382 in the Main Text, that model mechanistic differences and interannual variability are two of the sources that limit quantitative comparisons across source contribution studies.

Supplement Line 624 –

For dust, agriculture, transportation, and fires, agreement with previous national-level results were variable. For example, national-level fractional dust estimates *in 2017* were much *larger* for North Africa, the Middle East⁸, and China⁹, and *smaller* in India² *compared to previous studies*. *For India specifically, updates to the model deposition⁶ and dust size distribution schemes⁵, as well as interannual variability in dust emission fluxes and removal rates, likely contribute to the smaller total contribution from fine dust (< 2.5 μm diameter) in this work (~14.9%) relative to previous estimates (~38%) derived using an older version of the GEOS-Chem model*. As shown in SI-Fig. 5, *the model updates in this work improved the agreement with surface dust observations, however, the measured PWM dust concentrations at surface monitors were < 1 $\mu\text{g}/\text{m}^3$, indicating that current surface monitor locations may not provide an accurate characterization of the total population exposure to dust*. These uncertainties highlight the need *for increased monitoring and continued improvement to the model treatment of dust* to improve the accuracy of contribution estimates *from this source*.

c) I was wondering at what size range (Aitken/accumulation) the primary organic aerosols and black carbon are emitted in the model?

In this work, we follow the standard GEOS-Chem treatment of aerosol emission and hygroscopic growth. Primary organic and black carbon are both emitted primarily into the Aiken mode. We have provided additional details and references in the Supplemental SI-Text 3.

Supplement line 179 -

Relative humidity dependent aerosol size distributions and optical properties are based on the Global Aerosol Data Set^{10,11}, with updates for organics and secondary inorganics from observations^{12,13}, mineral dust^{5,14,15}, and absorbing brown carbon¹⁶.

d) I strongly suggest the authors to improve the representation of 95%CI ranges in the excess death estimates by combining the uncertainties in baseline disease rates and the CRFs.

In this work, we follow the approach of similar previous studies¹⁷⁻¹⁹ and report the 95% confidence interval in PM_{2.5} disease burden estimates, calculated from uncertainties in parameters associated with the concentration response functions (CRFs). In response to an additional comment from Reviewer #2, we now also report these confidence intervals for the attributable mortality estimates throughout the Main Text and Supplement (see our full response below). However, we agree with the Reviewer that additional uncertainty is present from uncertainties in the baseline mortality data and therefore have followed the approach of Achakulwisut, et al.¹⁹ and provided an additional sensitivity study to test the uncertainty in the disease burden estimates associated with uncertainty in the baseline mortality data. These results are discussed in the new SI-Text 7. A new SI-Fig. 7 also shows the 95% Confidence Intervals for attributable mortality estimates for 21 world regions when calculated from the uncertainties in the CRFs and the baseline data. SI-Fig. 7 shows that for most regions, the uncertainty ranges associated with the uncertainties in the CRFs are generally larger than those associated with the baseline mortality data.

Main Text – Line 466 –

*Following methods from previous similar studies¹⁷⁻¹⁹, the 95% CI of the 2017 PM_{2.5} disease burden is derived from uncertainties in the GBD2019 CRFs, resulting in a range of 2.72 million - 4.97 million global attributable deaths. An additional sensitivity study is presented in SI-Text 7 to test the impact of uncertainties associated with the baseline mortality data, which for the majority of world regions results in smaller uncertainty bounds than those associated with CRF uncertainties (SI-Fig. 7). ... In addition, sub-national fractional source contributions (Fig. 5) are limited to the resolution of the model and emissions, while the urban exposure estimates are further subject to *greater* uncertainties in the satellite-derived products for small spatial scales^{20,21}. Future developments of global high-resolution simulations, as well as increasing the accuracy and precision of satellite-derived PM_{2.5} estimates will serve to reduce these uncertainties in PM_{2.5} mass and source contributions at both the national and sub-national scales.*

Supplemental - Line 728 –

SI-Text 7. Uncertainty Sensitivity Study

We conduct an additional sensitivity test to account for potential uncertainties in the PM_{2.5} disease burden associated with the age- and disease-specific baseline mortality data from the 2019 GBD. The 95% uncertainty ranges are calculated by applying lower and upper estimates of the baseline mortality data to the PAF in Equation 2, derived from the mean CRF. The resulting 95% confidence intervals for 21 world regions are shown in SI-Fig. 7, compared to the 95% CIs derived from uncertainties in the

mean CRFs (reported in the Main Text). As the upper and lower limits in the baseline and CRF datasets are both estimated from multiple draws of underlying distributions, propagating the uncertainties from these two input variables likely leads to an overestimate in the 95% CI for the total attributable disease burden. SI-Fig. 7 shows that for most regions, the 95% CI associated with uncertainties in the CRFs encompass the 95% CIs associated with uncertainties in the baseline mortality estimates. Additional uncertainties in the PM_{2.5} exposure estimates and modeled fractional source contributions are not considered here to due computational limitations.

SI-Fig. 7. Total disease burden estimates and confidence intervals for 21 world regions, derived from uncertainties in CRFs and baseline mortality data. Total disease burden estimates are from Data File 1. Uncertainty ranges illustrate the 95% CI derived from uncertainty estimates in the CRFs (blue) and baseline mortality data (red). The bounds for South and East Asia are shown on an expanded scale to the right.

Review #2

This manuscript analyzed global health issues by PM_{2.5} with a broader view of source sector- and fuel-specific contributions in 2017. The manuscript also forecast the avoided deaths in certain PM_{2.5} control scenario. The topic of health effect of global PM_{2.5} is relevant to the scope of Nature Communications, but there are some statements to be clarified. I would recommend this manuscript to be accepted with minor modification.

We thank the Reviewer for their detailed comments and questions, which have helped improve the overall quality and clarity of the manuscript. We have addressed each comment below, including the addition of a new supplemental figure and consistent reporting of disease burden confidence intervals throughout the text.

Minor comments

1. Lines 43-44, the mortality should come with confidence interval. I think the CI is needed for the mortality appeared first time throughout the text.

We agree with the Reviewer and have added the 95% confidence interval at the suggested point in the abstract, as well as all other locations in the Main Text and Supplement where absolute mortality values (or neonatal incidences) are reported. In Response to Reviewer #1, we have also included an additional uncertainty sensitivity test in SI-Text 7. Please see the specific changes below and in our responses to Reviewer #1.

Abstract Line 7 –

Globally, 1.05 (95% Confidence Interval: 0.74-1.36) million deaths were avoidable in 2017 by eliminating fossil-fuel combustion (27.3% of the PM_{2.5} burden), with coal contributing to over half. Other dominant global sources included residential (19.2%; 0.74 [0.52-0.95] million deaths), industrial (11.7%; 0.45 [0.32-0.58] million deaths), and energy (10.2%; 0.39 [0.28- 0.51] million deaths).

Main Text

Line 125 –

Globally, we estimated 3.83 million deaths (95% Confidence Interval: 2.72-4.97 million) were attributable to annual ambient PM_{2.5} exposure in the year 2017 (Fig. 2: top left panel).

Line 130 –

In addition, there were a total of 2.07 (95% CI: 0.02 to 5.02) million attributable incidences of neonatal disorders (Low Birth Weight (LBW) + Pre-Term Births (PTB)) worldwide (Data File 1). ... The largest numbers of attributable deaths occurred in China (~1.4 (95% CI: 1.05-1.70) million) and India (0.87 (95% CI: 0.68-1.04) million), together accounting for 58% of the global total ambient PM_{2.5} mortality burden.

Line 156 –

No change was found in the global PWM PM_{2.5} concentration (Data File 3), however due to changes in population characteristics (i.e., size and age decomposition in a particular country), the attributable deaths increased from 3.8 (95% CI: 2.72-4.97) million to 4.1 (95% CI: 2.9-5.3) million in 2019 (consistent with GBD2019¹) (SI-Text 1; Data File 3).

Line 175 –

Results in Fig. 2 (and Data File 1) show that on the global scale, roughly 40% of the PM_{2.5} disease burden was attributable to residential (19.2%; 0.74 [95% CI: 0.52-0.95] million deaths), industrial (11.7%; 0.45 [0.32-0.58] million deaths), and energy (10.2%; 0.39 [0.28- 0.51] million deaths) sector emissions, which are typically associated with fuel combustion²². ... For example, Fig. 2 shows that nearly 1.05 (95% CI: 0.74-1.36) million or 27.3% of total PM_{2.5} attributable deaths could be avoided by

eliminating emissions from fossil-fuel combustion (coal = 14.1%, O&NG = 13.2%), with an additional 20% or nearly 0.77 (95% CI: 0.54-0.99) million deaths avoidable by eliminating solid biofuel combustion, primarily used for residential heating and cooking.

Line 205 –

This source was estimated to lead to 0.62 (95% CI: 0.44-0.80) million attributable deaths worldwide under the assumption of equal toxicity of all PM_{2.5} sources and components (Discussion). Other PM_{2.5} sources such as on-road transportation, non-combustion agriculture emissions, and anthropogenic dust each had relatively smaller global contributions ranging between 6.0% and 9.3% (0.23 [95% CI: 0.16-0.30] to 0.36 [0.25-0.46] million deaths).

Line 228 –

Of the three anthropogenic fuel categories (coal, O&NG, and solid biofuel), coal was the largest source of PM_{2.5} attributable mortality in China (22.7%; 315,000 [95% CI: 239,000-385,000] deaths), O&NG was the largest contributor in Egypt, Russia, and the United States (13.7%-27.9%; 9,000 [4,000-16,000] to 13,000 [4,500-24,000] deaths), and solid biofuel combustion was largest (12.3%-36.0%; 6,000 [4,500-8,000] to 250,000 [196,500-300,000] deaths) in the remaining five countries.

Line 274 –

At the national-level, South Africa and neighboring Eswatini both had the largest relative coal contributions of all countries at more than 36.5% each (~9,000 [95% CI: 6,000-12,500] deaths total).

Line 281 –

Of all world regions, North America and Western Europe had the largest relative O&NG contributions at ~25% each (43,000 [95% CI: 19,500-72,500] deaths total), while the lowest was in Central Sub-Saharan Africa at 2.5% (less than 1,000 deaths total). Third, regional solid biofuel contributions (largely from the residential sector) were largest in South and Southeast Asia at between 29.2%-31.2% each (373,500 [95% CI: 279,500-465,000] deaths total). ... National-level fractional contributions ranged from 0.2% in small island nations to at least 40% in Guatemala, Nepal, and Rwanda (8,500 [95% CI: 6,500-11,000] total deaths).

Line 400 –

The use of solid biofuel across all sectors in 2017 contributed to an additional 767,000 (95% CI: 543,000-994,500) deaths worldwide (20%), with this source in India and China again responsible for roughly 11% of the global PM_{2.5} disease burden.

Line 437 –

The number of global attributable deaths from the waste sector (184,000; 95% CI: 130,500-238,500 deaths) in 2017 was 30% lower than the only previous estimate of domestic waste burning²³. ... For international shipping, global mortality estimates (27,000; 95% CI: 19,000-35,000 deaths) fell within the range of a previous 2002 estimate²⁴, but were 75-95% lower than a more recent study, largely due to differences in the CRFs²⁵.

Supplement

Line 77 –

For neonatal disorders, the incidence associated with outdoor PM_{2.5} exposure totaled to 2.07 (95% CI: 0.02-5.02) million worldwide, which increased marginally to 2.09 (95% CI: 0.02-5.06) million incidences in 2019.

Line 131 –

Globally, the updated GEMM CRFs estimated a PM_{2.5} attributable burden of 6.2 (95% CI: 4.4 to 7.8) million deaths in 2017.

Line 623–

For 2017, we estimate a total of nearly 250,000 (95% CI: 189,500-305,500) deaths avoidable by eliminating both solid biofuel and coal use in the residential sector in China.

2. Lines 167-169, I am very curious with the higher mortality in China compared with India. I feel the CI for mortality is needed here. It is better to show the magnitude differences in population age distribution and the relative baselines associated with each disease between China and India. Also, any previous studies show the same conclusion?

Following the Reviewer's first suggestion, we have added the 95% CI for both the China and India mortality estimates at this location in the Main Text (below). The results from this analysis show that PM_{2.5}-attributable deaths are larger in China than in India. This result is not unique and is consistent with all previous studies that we are aware of that quantify the total attributable PM_{2.5} mortality for both China and India within the same study (e.g., GBD 2019 Risk Factor Collaborators ¹Lelieveld, et al. ¹⁷Lelieveld, et al. ¹⁸Burnett, et al. ²⁶).

As we note on line 135 in the Main Text, the “larger burden in China, despite a lower national PM_{2.5} exposure level reflects differences in population age distribution and the relative baselines associated with each disease in each country.” Following the Reviewer's suggestion to more clearly demonstrate this point, we have added a new Supplemental Figure (SI-Fig. 1) that illustrates the (A) disease- and age-specific baseline mortalities in China and India, the (B) GBD2019 CRFs with the PM_{2.5} exposure levels in each country, and (C) the resulting absolute number of disease-specific PM_{2.5} attributable deaths resulting from the baseline mortalities and CRFs in each country. This figure illustrates that despite a lower PM_{2.5} exposure level (49.8 vs 80.2 μg m⁻³), the baseline mortality in China is much larger than in India, particularly for Stroke, IHD, and Lung Cancer, which results in a larger number of total deaths in China from these diseases despite the lower exposure. In contrast, the combined larger number of LRI mortalities in India, combined with higher exposure, result in a larger number of LRI deaths in India than in China. We have added a reference to this figure on Line 138 in the Main Text.

Line 132 –

National-level results for 204 countries are provided in the center map of Fig. 2 (*and Data File 1*). The largest numbers of attributable deaths occurred in China (~1.4 [95% CI: 1.05-1.70] million) and India (0.87 [95% CI: 0.68-1.04] million), together accounting for 58% of the global total ambient PM_{2.5} mortality burden. The larger burden in China, despite a lower national PM_{2.5} exposure level reflects differences in population age distribution and the relative baselines associated with each disease in each country (*SI-Fig. 1*).

SI-Fig. 1. PM_{2.5} Disease Burden Comparison for India and China. (A) GBD2019 background baseline mortality data as a function of population age and disease. (B) GBD2019 concentration response functions (same as SI-Fig. 2), with dashed lines showing PWM PM_{2.5} concentrations for China and India. (C) Total PM_{2.5} attributable deaths in China and India as a function of disease.

3. Line 176-178, is it a better way to project 2019 emission based on 2017 to meet the burden estimates for 2019? I understand this may bring additional uncertainty but at least a discussion is needed here.

In this analysis, we calculate fractional source contributions (sector and fuel-type) to PM_{2.5} mass using emission sensitivity simulations run with the GEOS-Chem model for the year 2017 (to match the latest emissions year). In an independent analysis step, we then calculate the total PM_{2.5} attributable disease burden for the year 2017 by applying 2017 PM_{2.5} exposure estimates (downscaled from exposure estimates used in the 2019 GBD) to the GBD2019 concentration response functions and year 2017 baseline mortality data. Source-specific disease burden contributions reported throughout the text are then calculated as the total PM_{2.5} disease burden for the year 2017, scaled by the modeled fractional source contributions from the emission sensitivity simulations.

As an additional sensitivity study, we also calculate the total PM_{2.5} attributable disease burden for the year 2019, using exposure estimates (reported in Data File 3) and baseline mortality data that are publicly available for the year 2019 (CRFs are independent of year). Therefore, emission projections between the years 2017 and 2019 are not required for the 2019 disease burden estimates as these are independent calculations from the model simulations. Source-specific disease burden contribution estimates for the year 2019 are outside the scope of this work, but could be estimated by applying the same modeled fractional source contributions from the 2017 emission sensitivity simulations to the 2019 disease burden estimates.

We have edited the first sentence in this paragraph and added a sentence to SI-Text 1 to better clarify that the disease burden estimates are independent from the model emission sensitivity simulations and do not require emission projections.

Main Text lines 154 -

As an additional sensitivity test, exposure and burden estimates for the year 2019 were additionally calculated with publicly available 2019 exposure estimates and national-level baseline burden data (SI-Text 1).

Supplemental Line 66 –

Following the downscaling procedure described in the Methods (and SI-Text 9), we apply high-resolution (gridded at ~1 km × ~1km) exposure estimates for the year 2019 (weighted by 2019 gridded population²⁷) to the GBD2019 CRFs with 2019 baseline mortality data to assess changes in the estimated disease burden between 2017 and 2019. Disease burden estimates are independent from model emission sensitivity simulations and do not require changes or projections in emissions.

4. Lines 536-539, the validation of geographical-weighted regression between AOD and PM_{2.5} is needed here at least with a summary in SI although the cited paper discussed.

In this analysis, we use publicly available high-resolution gridded estimates of PM_{2.5} surface concentrations to estimate PM_{2.5} exposure levels. As described in the Methods, we use two publicly available PM_{2.5} data products for the exposure estimates, the 0.1°×0.1° (~10 km × 10 km) PM_{2.5} concentrations used in the GBD2019 analysis¹, as well as 0.01°×0.01° (~1 km × 1 km) PM_{2.5} concentration estimates from Hammer et al., 2020²⁰. To maintain consistency in this work with GBD exposure estimates, we incorporate the increased spatial information from Hammer et al.²⁰ by spatially downscaling the GBD estimates, as described in SI-Text 10 and illustrated in SI-Fig. 9. The development and validation of both public datasets are described in previous publications^{20,21,28}. In this manuscript, we have provided an evaluation in the supplement (SI-Fig. 10) of the GEOS-Chem simulation, ~10 km × 10 km GBD estimates, and ~1 km × 1 km Hammer estimates relative to 2017 surface PM_{2.5} observations.

As we are using publicly available PM_{2.5} datasets, it is outside the scope of this work to validate the geographically weighted regression (GWR) that is used to develop the Hammer et al.,²⁰ PM_{2.5} product. As described in Hammer, et al.,²⁰ geophysical estimates of PM_{2.5} surface concentrations are first derived using satellite AOD and the GEOS-Chem model to represent the geophysical relationship between PM_{2.5} mass and AOD. A geographically weighted regression is then used to statistically fuse the geophysical PM_{2.5} estimates to available ground PM_{2.5} observations. Figs. 3 and 6 in Hammer et al.,²⁰ show that in a comparison with ground observations, the correlation slope and coefficient increase from 0.9 (slope) and 0.81 (r²) to 0.91 (slope) and 0.92 (r²) as a result of the GWR process.

We have edited this section of the Methods to better clarify and reflect that we are using publicly available datasets that are described and evaluated elsewhere. Therefore, we include references to these datasets and re-focus the text on aspect of the analysis that were conducted specifically for this work.

Line 537 –

To maintain consistency with the GBD project, *while also improving the accuracy of the population-exposure estimates, we downscale the 2019 GBD exposure estimates^{1,21,28} to a 0.01°×0.01° (~1 km × 1 km) grid using a newly available high-resolution PM_{2.5} dataset from Hammer, et al.²⁰. SI-Text 10 (SI-Fig. 9) describes this process of spatial downscaling by incorporating the spatial information from the Hammer, et al.²⁰ product. This downscaling process is independent of the modeled fractional source contribution results and maintains the average PM_{2.5} mass concentration (area average only) from the original GBD product. The sensitivity of the PM_{2.5} exposure estimates to the downscaling process are evaluated in SI-Text 10 and SI-Fig. 10. Exposure estimates for the year 2019 were derived using these same methods with both GBD and Hammer, et al.²⁰ data for the year 2019.*

Review #3

The MS Source Sector and Fuel Contributions to Ambient PM_{2.5} and Attributable Mortality Across Multiple Spatial Scales is overall very well written and present a comprehensive and novel global assessment of sectoral contributions to PM_{2.5} concentrations at different spatial scales.

I only have a few comments and suggestions for minor changes before I would consider this MS suitable for publication:

We thank the Reviewer for their positive and detailed comments and suggestions for improvements. We have responded to each comment and question below. Most notably, we have made the requested updates to Figs. 1, 3, and SI-Figs. 6 and 7. We have also edited the introduction to more clearly highlight the benefit of a global study such as ours, have clarified the extent to which indoor pollution sources are considered in this work, and have clarified the importance of SOA to PM_{2.5} mass and attributable mortality in specific regions.

1. L50: "responsible for 4.1 million deaths" - throughout the MS, you primarily focus on association with premature deaths/mortality, and I would suggest to nuance this statement accordingly here.

We agree with the Reviewer that the noted line requires more nuance. To maintain consistency with the terminology used in the Global Burden of Disease, we have clarified that these are 'attributable' deaths. We have edited this terminology where necessary for consistency throughout the Main Text and Supplement.

Main Text Line 3 –

Long-term exposure to ambient (outdoor) fine particulate matter less than 2.5 micrometers in diameter (PM_{2.5}) is the largest environmental risk factor for human health, *with an* estimated 4.1 million *attributable* deaths *worldwide* (7.3% of the total number of global deaths) in 2019¹.

2. L51/52: I understand your focus is on the contribution of fossil fuels, but a couple of sentences at least here (or in methodology) on the potential contribution by organic aerosol would be useful. I am not suggesting a major edit, but identifying that composition of PM_{2.5} is still uncertain in particular in relation to the contribution of SOA. I would argue that an in depth discussion of condensable fractions and detailed composition is beyond the scope of your global assessment, however. This is linked as well to L350 where you make the point of highlighting the importance of developing region-specific AQ strategies.

On (original) lines 51-52, we state that "Outdoor PM_{2.5} mass is primarily composed of inorganic ions, carbonaceous compounds, and mineral dust." In this context, carbonaceous compounds refer to black and organic carbon (both primary and secondary) containing aerosol. We have edited this sentence to clarify this detail. In addition, we also note in the GEOS-Chem description section in SI-Text 3 that the model includes both black and organic carbon aerosol and have clarified the details of the direct yield scheme in GEOS-Chem for secondary organic aerosol (SOA). As the Reviewer suggests, we have also added a clarifying statement in the (original) Line 359 to highlight that modeled SOA concentrations are uncertain, but that they can be a dominant source of PM_{2.5} mass at the sub-national scale.

Main Text –

Line 6 –

Outdoor PM_{2.5} mass is primarily composed of inorganic ions, carbonaceous compounds (*black and organic carbon, including secondary organic aerosol*), and mineral dust.

Line 340–

Similarly, while the transportation sector was the largest PWM PM_{2.5} source in the U.S., Fig. 5c illustrates regionally varying sources, with dominant contributions from forest fires in the west, windblown dust in the arid southwest, agricultural, on-road transportation, and energy throughout the midwest and east coast, and *highly uncertain sources such as secondary organic aerosol (SOA)* in the southeast.

Supplemental Section - Line 184 –

We use the simple, irreversible, direct yield scheme for *secondary organic aerosol (SOA)* from Kim, et al.²⁹, as this mechanism has been shown to better reproduce available observations of global organic aerosol mass relative to the more complex scheme³⁰.

3. L81ff: I agree with your assessment that there are not many global studies, but it is not immediately clear what the global studies provide over regional or national scale studies. The key issue of transboundary effects of e.g. SIA and in particular due to ammonium nitrates and -sulfates is clear and e.g. addressed by the UNECE CLRTAP or TFTAP. Would the key argument for global studies be that such an assessment may help to avoid pollution transfer through identifying the contribution of energy-intensive industrial production to other world regions? What I am after is a brief argument in the introduction to put the global study in perspective - not just that there are few, but that they can play a vital role in international policy design.

We appreciate this suggestion from the Reviewer and agree that it is important to highlight the key benefits of a global air pollution study. These largely include the global coverage and consistency in methods and sectoral definitions that allow us to place air pollution in a global context, while also allowing for regional comparisons. As the Reviewer also notes, the nature of global simulations allows such studies to also account for the transport and impact of air pollution across political boundaries. To more clearly highlight the benefit of a global study, we have provided the following edits in the introduction.

Line 17-

As air pollution and atmospheric chemistry do not adhere to political boundaries³¹⁻³³, *mitigation efforts require consideration of transboundary effects across multiple locations, informed by studies of PM_{2.5} source contributions and the attributable disease burden across a range of sub-national to global scales.*

Line 22 –

Source contribution studies *across multiple spatial scales* help to inform specific mitigation strategies and prioritize limited resources for effective action³⁴.

Line 30 –

Therefore, to assess the *global and regional PM_{2.5} disease burden and its source contributions*, recent studies have employed 3D chemical transport models as a means to relate changes in surface emissions to atmospheric PM_{2.5} concentrations.

Line 48 –

In contrast, global-scale studies that account for transboundary effects using both consistent methodologies and sectoral definitions across all world regions help to place air pollution in a global context and allow for comparability of the disease burden and its source contributions across multiple locations. Relatively few of these previous global studies, however, have provided an assessment of the contributions from more than one source sector or aggregate fuel category in recent years^{17,18,35-37}, thereby limiting their ability to inform or prioritize specific air quality management policies under current global conditions.

Line 71 - Third, to capture *and compare* national and sub-national impacts across all world regions, these studies additionally require high-resolution PM_{2.5} exposure estimates, such as those that utilize recent advances in satellite retrievals, chemical transport models, and ground-based monitoring²⁰. Lastly, integration of these source simulations and exposure estimates with updated disease-specific CRFs can motivate policy action by refining previous PM_{2.5} disease burden estimates^{18,26,35,38}, incorporating spatial variation in the underlying health status and cause of death composition, and by *comparably* quantifying the dominant sources of this burden across global, national, and sub-national scales.

4. Fig 1. This is a key output and panel C in particular is fairly dense in that it presents a lot of data dimensions. For the scatterplot, I am wondering if the shapes of observations sites (and thus the legend) could be omitted here and presented in the SI, as a key interesting element is the regional clustering, which could be drawn out better here. Either that, or increasing the size of Panel C to a full-width figure may much improve accessibility to readers.

To better highlight the regional clustering of data points, we have followed the Reviewer's suggestion and have adjusted Figure 1 so that Panel C is now the full width of the figure. In addition, all observational data points are collected from publicly available data sources, which have been described in extensive detail in Supplemental Section SI-Text 4.

Figure 1. legend remains the same

5. Fig 2. Similar comments on the colouring and shading of the pie charts - It becomes fairly difficult for a reader to fully appraise the different data content where colour/shading is very similar. There is without doubt a trade-off between density of information presented and accessibility by readers here.

We have developed Fig. 2 as a summary ‘snapshot’ figure to illustrate the extent of information provided in this analysis ($PM_{2.5}$ exposure and disease burden estimates and fractional sector, fuel type, and disease contributions at global, regional, national, and sub-national scales). As the Reviewer notes, in attempting to highlight this information, there is a trade-off between information density and ease of readability. As the fractional contributions from a large number of individual source sectors and fuel types are the main novelty of this work, as well as the extent of the global converge of our analysis (204 countries), we have chosen to display the full extent of this information in Fig. 2, but only for the global region and for a select number (9) of countries. We agree with the Reviewer that the large number of source sectors, fuels, and diseases may make it difficult to quantitatively discern the smaller sources in each pie chart.

Therefore, we have also provided complete quantitative results for fractional source, fuel, and disease contributions, total disease burden estimates, and PWM PM_{2.5} concentrations for each of the 21 regions, 204 countries, and 200 sub-national areas in Supplemental Data Files 1 and 2. We make a specific note of these Data Files in the figure caption for Fig 2. We have also edited the text to further highlight that these data are provided in Supplemental Data Files 1 and 2. Further, we have also prepared an online data visualization platform that is built on top of the manuscript Supplementary Data Files. We would be happy to include a link to this tool in the manuscript, if the Editor deems this appropriate.

Line 131 –

National-level results for 204 countries are provided in the center map of Fig. 2 (*and Data File 1*).

Line 176 –

Results in Fig. 2 (*and Data File 1*) show that on the global scale, roughly 40% of the PM_{2.5} disease burden was attributable to residential (19.2%; 0.74 [95% CI: 0.52-0.95] million deaths), industrial (11.7%; 0.45 [0.32-0.58] million deaths), and energy (10.2%; 0.39 [0.28-0.51] million deaths) sector emissions, which are typically associated with fuel combustion²².

6. L227/8: the assumption of equal toxicity is appropriate here and I would challenge that at all; I am wondering if the rather different world regions and PM_{2.5} compositions assessed, however, could yield some insights if differences in premature mortality associated with different composition 'fingerprints' indicate and more or less 'toxic' mixes? Or are confounding factors too complex to draw out such a conclusion?

As the Reviewer notes, the many confounding factors such as the population age structure and baseline mortalities are too complex to draw out the potential relationship between total PM_{2.5} attributable deaths and the chemical composition of PM_{2.5} at exposure-relevant scales. As this work focuses on quantifying policy-relevant emission sector contributions, a detailed composition-focused analysis is outside the scope of this work.

7. Fig 3. numbers in legends are very difficult to read and could benefit from revision; in addition, the largest number of attributable deaths seems to not be associated with a sized circle in Panel B?

We have followed the Reviewer's suggestion and increased the font size of the attributable deaths and PWM PM_{2.5} concentrations in Figure 3 and the similar supplemental SI-Fig. 3.

To address the Reviewer's second question, we purposefully omitted a sized circle in panel B for the total global deaths so that the circles would still be visible for other regions. To clarify this purposeful omission, we have made a note of this in the figure legend (and the legend of SI-Fig. 3).

Fig. 3 Relative (fractional) source and fuel contributions to annual population-weighted *mean* PM_{2.5} mass and attributable deaths. Panels (A, C): Normalized sectoral source contributions for 21 world regions and the global average (A) and top 20 countries (C). Sorted by decreasing number of ambient PM_{2.5} attributable deaths (rounded to the nearest 1000). Panels (B, D): Normalized contributions from the combustion of three fuel categories and remaining PM_{2.5} sources. To the right of panels B and D, annual *PWM* PM_{2.5} concentrations and associated attributable deaths are provided for each region/country. Relative amounts are illustrated by relative dot sizes (*except for the global total disease burden*). Concentrations above or equal to the global average are colored red.

8. L394/6: What role do population density play in the occurrence of the largest number of PM_{2.5} attributable deaths in China and India? For instance, programmes in India to reduce indoor burning of solid wood or waste for cooking in deprived areas and informal settlements in Africa by introducing natural gas burners may reduce premature mortality by improving indoor air quality, but could be considered negative for outdoor air quality due to being related to fossil fuel combustion. Either here or somewhere, it could be helpful to state if indoor AQ is at all considered, or not?

To address the Reviewer's first question, gridded population data are used in this analysis to calculate the population-weighted annual mean PM_{2.5} mass concentrations, which are used as estimates of population exposure in the disease burden analysis. The Reviewer is correct in that the large populations (and corresponding baseline mortalities) in China and India relative to other locations contribute to the large PM_{2.5} attributable disease burden in these locations.

To address the Reviewer's second comment, this work considers indoor sources of air pollution (such as residential sector indoor cooking and heating) to the extent that these sources impact ambient

PM_{2.5} concentrations. For example, the residential sector emissions in the CEDS emissions dataset (used as input to the GEOS-Chem model) are largely associated with household cooking and heating activities. We have added clarification for this point in this section, as well as in the Methods (see below).

Lastly, we have also added an additional sentence in this section to highlight that net air quality improvements may be realized in some instance of switching from biomass to fossil fuel energy sources.

Line 402 –

Solid biofuel emissions in countries throughout South and Southeast Asia, as well as Central and Western Sub-Saharan Africa were largely associated with residential solid biofuel use for *household* heating and cooking (Fig. 5b).

Line 416 –

Additional considerations of net air quality benefits will also be important in regions where a transition from residential solid biofuel use to fossil fuel energy sources may lead to immediate indoor and outdoor air quality improvements³⁹, while at the same time increasing the relative fossil fuel contributions.

Line 628–

As a result of these adjustments, the PM_{2.5} attributable mortality and source contribution results presented in this analysis reflect contributions from indoor sources of air pollution (e.g., biomass combustion for residential heating and cooking) to the extent that they impact ambient PM_{2.5} concentrations.

9. SI-Fig6/7. These figures and schematics try to provide an overview of the flow of data and how they are combined, which is much appreciated. I would, however, argue that they are not as clear and accessible as they could be. The use of GBD, EO and ACTM data, alongside other data sources, is indispensable, but it comes at a cost as it is not fully clear and transparent of how these sources are combined and how e.g. inherent uncertainties or ranges of data are accounted for. I would as well suggest to include a preliminary step in particular on the GEOS-CHEM modelling, as emission data inputs seem to be missing in SI-Fig 6. While I do not have a perfect solution for a clearer flowchart either, perhaps removing the maps as examples in SI-Fig 6 and instead focusing on more descriptive text or representation of the actual data types (and their sources) which are used in each stage could be a solution?

We appreciate the Reviewer's feedback on the clarity of these schematic figures. We have updated the flow chart in SI-Fig. 6 (now SI-Fig. 8) as recommended by the Reviewer to illustrate the preliminary step of the emissions input into the GEOS-Chem modeling. We have retained the maps for illustrative purposes but have reorganized the flow chart to simplify the illustration of how the input datasets and results are used throughout the analysis. We have also updated the descriptive text in SI-Text 9 to better describe the simplified schematic and the data that are used in each step. To further increase the transparency and reproducibility of our analysis, we have also provided a complete analysis scripts package at: <https://github.com/emcduffie/GBD-MAPS-Global>.

Similarly, we have also updated SI-Fig. 7 (now SI-Fig. 9) to more clearly illustrate an example of the downscaling procedure. As with the core analysis scripts, the publicly available scripts package also includes the code for the downscaling procedure for increased transparency and reproducibility.

SI-Fig. 8: Overall methodological workflow schematic. *The relevant equations and data from SI-Text 9 are indicated in each step.*

Updates to SI-Text 9:

Line 772 –

In Step 1, gridded global emissions of PM_{2.5} precursors are developed as a function of source sector and fuel-type (SI-Table 5; anthropogenic emissions largely from the CEDS_{GBD-MAPS} inventory), as described in McDuffie, et al. ²². In Step 2, emissions are used as input in an updated version of the GEOS-Chem 3D chemical transport model (described in SI-Text 3), with the simulated PM_{2.5} concentrations validated against available mass and composition surface observations (described in SI-Text 4). In Step 3, a series of zero-out emission sensitivity simulations are conducted (SI-Table 2) with the GEOS-Chem model and emission inputs. The resulting PM_{2.5} concentrations from each simulation are compared to the base simulation (with all emission sources) to quantify the modeled fractional PM_{2.5} contributions (reported in Data Files 1 (sectors) and 2 (fuel-types)). In Step 4, high-resolution PM_{2.5} exposure estimates are derived by downscaling exposure estimates from the 2019 GBD (described in SI-Text 10; reported in Data Files 1 and 2) and are applied to the fractional model source contributions from Step 2 to quantify absolute source-specific contributions to ambient PM_{2.5} mass. In Step 5, CRFs from the GBD2019 (SI-Fig. 2) are combined with downscaled PM_{2.5} exposure estimates from Step 4 to calculate the total ambient PM_{2.5} disease burden (reported in Data Files 1 and 2). The total burden is combined with modeled fractional source contributions from Step 3 to calculate source-specific burden contributions reported throughout the manuscript. Lastly, Step 6 highlights the data assets that are associated with this analysis and manuscript, including the analysis scripts, model source code, input emissions, CRFs, baseline burden and exposure estimate datasets (<https://github.com/emcduffie/GBD-MAPS-Global>), and the global, regional, national, and subnational source sector and fuel contribution results (Data Files 1 and 2).

SI-Fig. 9. Simplified schematic of the spatial downscaling procedure. Values in each example grid box represent example $PM_{2.5}$ mass concentrations in units of $\mu g m^{-3}$. In actuality, one of the $0.1^\circ \times 0.1^\circ$ grid boxes in Step 1 above corresponds to 100 grid boxes of $0.01^\circ \times 0.01^\circ$ resolution, not the four as shown here. In this figure ‘GWR’ refers to the high-resolution $PM_{2.5}$ estimates from Hammer, et al. ²⁰.

I would like to stress again that all comments are of a minor and predominantly presentational/editorial in nature. The methodology and robustness of the analysis are sound and the degree of detail presented - while it could at times be clearer - appropriate for a study of this caliber. I have very much enjoyed reading the MS and look forward to see it published hopefully soon.

We thank the Review again for their very insightful and helpful comments.

Review #4

The title and abstract of this manuscript accurately describe the work presented. This work uses highly spatially resolved (for a global domain) modelled data of speciated PM_{2.5} concentrations and PM_{2.5}-disease pair concentration response functions (CRFs) from the Global Burden of Disease to calculate spatially-resolved numbers of attributable premature deaths across the globe. This in itself is not novel. The novelty is the assignment of attributable premature deaths to each of a series of source sectors contributing to PM_{2.5} (via primary and secondary PM_{2.5}) and, separately, the assignment of attributable premature deaths to a series of fuel types underpinning the emissions. A further novelty is the quantification of these source and fuel assignments at multiple scales of area disaggregation, from GBD region, to individual country, to sub-national and urban spatial scales.

The manuscript is extremely well presented. The description is technically focused and unambiguously written. A lot of information is presented in each figure. A comprehensive supplementary information is provided which includes further methodological descriptions and results and discussion of sensitivity studies. Data files are also provided.

The work is of very high quality, provides a lot of interesting and insightful data and is of global interest. The discussion is supported by the methods and the data presented.

The results presented do, of course, critically depend on the quality of the contributing datasets and of the atmospheric chemistry modelling. The team work at the forefront of this research area. Many of the methodological approaches, and much of the data underpinning this study, are related to the GBD framework so that the work presented here is therefore consistent with GBD output. This applies both to the PM_{2.5} modelling and to the CRFs and underlying mortality data applied. The work builds on the team's previous work and the paper is comprehensively referenced. The derivation of sector contributions to PM_{2.5} uses the zero-out or so-called brute force modelling approach. This uses model runs in which there is complete removal of all emissions from a given source. This is widely used. However, it does mean that the sum of the changes in PM_{2.5} arising from elimination of each source individually exceeds the PM_{2.5} concentration in the base simulation. This work uses the standard approach of calculating the individual source sensitivity relative to the sum of the effects from all the sources rather than relative to the baseline PM_{2.5} concentration. A more fundamental issue in this brute-force approach is that the relative contribution assigned to each source may be different to the relative contributions derived from model simulations with more moderate emissions perturbations or when multiple sources are perturbed together. The authors of this paper recognise and highlight these issues. My view is that the approach taken here is satisfactory and that the general trends revealed in this study are likely to be fine, as long as the specific numbers aren't taken too literally as absolutely correct. And the output of this study is surely much more sensitive to uncertainty in the model input emissions than to inaccuracies introduced by the brute-force approach.

I support publication of this work essentially as it is. I make only a couple of minor observations and draw attention to one presentational error.

We thank the Reviewer for their positive and thoughtful comments on our analysis approach and manuscript presentation. We appreciate their attention to detail and have provided the clarifications requested below and have fixed the error in our original reference list.

1. L41: Insert the words 'annually' to read "Nearly 1.05 million deaths annually worldwide....."

We have added the phase ‘in 2017’ to this line in the abstract to clarify that our results are for a single year.

Abstract line 8 - *Globally, 1.05 (95% Confidence Interval: 0.74-1.36) million deaths were avoidable in 2017 by eliminating fossil-fuel combustion (27.3% of the PM_{2.5} burden), with coal contributing to over half*

2. Fig. 5: Maps that highlight the largest contributor to something, in this case the sector with the highest contribution to mortality, can be misleading because of the potential very different absolute contributions of the highest contributing source in different places. For example the highest contributing source might contribute 40% in one place but only 10% somewhere else. This does not make the maps less useful, but a caveat to this effect could be included in the discussion of the figure.

We agree that while the maps in Fig. 5 provide spatial information on the dominant source sectors, they provide limited information about the magnitude of these source contributions relative to other sources and include no information about the absolute contributions. This is one reason why we have also included the pie charts for select sub-national regions in Fig. 5, as a way to illustrate the full set of contributions from all sources (not just the dominant source). We have added the following sentence to the Main Text to highlight this point, that in many regions, a large number of sources collectively contribute to PM_{2.5} mass and exposure.

Main Text line 351 –

Pie charts in Fig. 5, however also highlight that in all regions, a large number of sources collectively contribute to sub-national PM_{2.5} mass formation, not only the largest sources illustrated in the map panels.

3. Reference #42 is the same as reference #6.

Thank you. We have replaced Reference #42 with Reference #6 and have updated this reference in the Supplemental material as well.

Response Document References

- 1 GBD 2019 Risk Factor Collaborators. Global burden of 87 risk factors in 204 countries and territories, 1990–2019: a systematic analysis for the Global Burden of Disease Study 2019. *The Lancet* **396**, 1223-1249, doi:[https://doi.org/10.1016/S0140-6736\(20\)30752-2](https://doi.org/10.1016/S0140-6736(20)30752-2) (2020).
- 2 GBD MAPS Working Group. *Burden of Disease Attributable to Major Air Pollution Sources in India. Special Report 21.*, (Health Effects Institute [Available at: <https://www.healtheffects.org/publication/gbd-air-pollution-india>], 2018).
- 3 Fairlie, D. T., Jacob, D. J. & Park, R. J. The impact of transpacific transport of mineral dust in the United States. *Atmos. Environ.* **41**, 1251-1266, doi:<https://doi.org/10.1016/j.atmosenv.2006.09.048> (2007).
- 4 Philip, S. *et al.* Anthropogenic fugitive, combustion and industrial dust is a significant, underrepresented fine particulate matter source in global atmospheric models. *Environmental Research Letters* **12**, 044018, doi:10.1088/1748-9326/aa65a4 (2017).
- 5 Zhang, L., Kok, J. F., Henze, D. K., Li, Q. & Zhao, C. Improving simulations of fine dust surface concentrations over the western United States by optimizing the particle size distribution. *Geophysical Research Letters* **40**, 3270-3275, doi:<https://doi.org/10.1002/grl.50591> (2013).
- 6 Luo, G., Yu, F. & Schwab, J. Revised treatment of wet scavenging processes dramatically improves GEOS-Chem 12.0.0 simulations of nitric acid, nitrate, and ammonium over the United States. *Geosci. Model Dev. Discuss.* **2019**, 1-18, doi:10.5194/gmd-2019-58 (2019).
- 7 Li, C. *et al.* Trends in Chemical Composition of Global and Regional Population-Weighted Fine Particulate Matter Estimated for 25 Years. *Environ. Sci. Technol.* **51**, 11185-11195, doi:10.1021/acs.est.7b02530 (2017).
- 8 Giannadaki, D., Pozzer, A. & Lelieveld, J. Modeled global effects of airborne desert dust on air quality and premature mortality. *Atmos. Chem. Phys.* **14**, 957-968, doi:10.5194/acp-14-957-2014 (2014).
- 9 Hu, J. *et al.* Premature Mortality Attributable to Particulate Matter in China: Source Contributions and Responses to Reductions. *Environ. Sci. Technol.* **51**, 9950-9959, doi:10.1021/acs.est.7b03193 (2017).
- 10 Martin, R. V., Jacob, D. J., Yantosca, R. M., Chin, M. & Ginoux, P. Global and regional decreases in tropospheric oxidants from photochemical effects of aerosols. *J. Geophys. Res. Atmos.* **108**, doi:<https://doi.org/10.1029/2002JD002622> (2003).
- 11 Koepke, P., Hess, M., Schult, I. & Shettle, E. P. Global Aerosol Dataset. (Max-Planck Institute for Meteorology, Hamburg, Germany, 1997).
- 12 Drury, E. *et al.* Synthesis of satellite (MODIS), aircraft (ICARTT), and surface (IMPROVE, EPA-AQS, AERONET) aerosol observations over eastern North America to improve MODIS aerosol retrievals and constrain surface aerosol concentrations and sources. *J. Geophys. Res. Atmos.* **115**, doi:<https://doi.org/10.1029/2009JD012629> (2010).
- 13 Latimer, R. N. C. & Martin, R. V. Interpretation of measured aerosol mass scattering efficiency over North America using a chemical transport model. *Atmos. Chem. Phys.* **19**, 2635-2653, doi:10.5194/acp-19-2635-2019 (2019).
- 14 Lee, C. *et al.* Retrieval of vertical columns of sulfur dioxide from SCIAMACHY and OMI: Air mass factor algorithm development, validation, and error analysis. *J. Geophys. Res. Atmos.* **114**, doi:<https://doi.org/10.1029/2009JD012123> (2009).
- 15 Ridley, D. A., Heald, C. L. & Ford, B. North African dust export and deposition: A satellite and model perspective. *J. Geophys. Res. Atmos.* **117**, doi:<https://doi.org/10.1029/2011JD016794> (2012).
- 16 Hammer, M. S. *et al.* Interpreting the ultraviolet aerosol index observed with the OMI satellite instrument to understand absorption by organic aerosols: implications for atmospheric oxidation and direct radiative effects. *Atmos. Chem. Phys.* **16**, 2507-2523, doi:10.5194/acp-16-2507-2016 (2016).

- 17 Lelieveld, J. *et al.* Effects of fossil fuel and total anthropogenic emission removal on public health and climate. *P. Natl. Acad. Sci.* **116**, 7192, doi:10.1073/pnas.1819989116 (2019).
- 18 Lelieveld, J., Evans, J. S., Fnais, M., Giannadaki, D. & Pozzer, A. The contribution of outdoor air pollution sources to premature mortality on a global scale. *Nature* **525**, 367, doi:10.1038/nature15371 (2015).
- 19 Achakulwisut, P., Brauer, M., Hystad, P. & Anenberg, S. C. Global, national, and urban burdens of paediatric asthma incidence attributable to ambient NO₂ pollution: estimates from global datasets. *The Lancet Planetary Health* **3**, e166-e178, doi:[https://doi.org/10.1016/S2542-5196\(19\)30046-4](https://doi.org/10.1016/S2542-5196(19)30046-4) (2019).
- 20 Hammer, M. S. *et al.* Global Estimates and Long-Term Trends of Fine Particulate Matter Concentrations (1998–2018). *Environ. Sci. Technol.*, doi:10.1021/acs.est.0c01764 (2020).
- 21 Shaddick, G. *et al.* Data Integration for the Assessment of Population Exposure to Ambient Air Pollution for Global Burden of Disease Assessment. *Environ. Sci. Technol.* **52**, 9069–9078, doi:10.1021/acs.est.8b02864 (2018).
- 22 McDuffie, E. E. *et al.* A global anthropogenic emission inventory of atmospheric pollutants from sector- and fuel-specific sources (1970–2017): an application of the Community Emissions Data System (CEDS). *Earth Syst. Sci. Data* **12**, 3413–3442, doi:10.5194/essd-12-3413-2020 (2020).
- 23 Kodros, J. K. *et al.* Global burden of mortalities due to chronic exposure to ambient PM 2.5 from open combustion of domestic waste. *Environmental Research Letters* **11**, 124022, doi:10.1088/1748-9326/11/12/124022 (2016).
- 24 Corbett, J. J. *et al.* Mortality from Ship Emissions: A Global Assessment. *Environ. Sci. Technol.* **41**, 8512–8518, doi:10.1021/es071686z (2007).
- 25 Sofiev, M. *et al.* Cleaner fuels for ships provide public health benefits with climate tradeoffs. *Nature Communications* **9**, 406, doi:10.1038/s41467-017-02774-9 (2018).
- 26 Burnett, R. *et al.* Global estimates of mortality associated with long-term exposure to outdoor fine particulate matter. *P. Natl. Acad. Sci.* **115**, 9592, doi:10.1073/pnas.1803222115 (2018).
- 27 CIESIN (Center for International Earth Science Information Network). Gridded Population of the World Version 4. (*Palisades, NY.*), doi:doi.org/10.1128/AAC.03728-14 (2017).
- 28 Shaddick, G. *et al.* Data integration model for air quality: a hierarchical approach to the global estimation of exposures to ambient air pollution. *J. Roy. Stat. Soc. C-App.* **67**, 231–253, doi:10.1111/rssc.12227 (2018).
- 29 Kim, P. S. *et al.* Sources, seasonality, and trends of southeast US aerosol: an integrated analysis of surface, aircraft, and satellite observations with the GEOS-Chem chemical transport model. *Atmos. Chem. Phys.* **15**, 10411–10433, doi:10.5194/acp-15-10411-2015 (2015).
- 30 Pai, S. J. *et al.* An evaluation of global organic aerosol schemes using airborne observations. *Atmos. Chem. Phys. Discuss.* **2019**, 1–39, doi:10.5194/acp-2019-331 (2019).
- 31 Liang, C. K. *et al.* HTAP2 multi-model estimates of premature human mortality due to intercontinental transport of air pollution and emission sectors. *Atmos. Chem. Phys.* **18**, 10497–10520, doi:10.5194/acp-18-10497-2018 (2018).
- 32 Zhang, Q. *et al.* Transboundary health impacts of transported global air pollution and international trade. *Nature* **543**, 705–709, doi:10.1038/nature21712 (2017).
- 33 Meng, J. *et al.* Source Contributions to Ambient Fine Particulate Matter for Canada. *Environ. Sci. Technol.* **53**, 10269–10278, doi:10.1021/acs.est.9b02461 (2019).
- 34 West, J. J. *et al.* “What We Breathe Impacts Our Health: Improving Understanding of the Link between Air Pollution and Health”. *Environ. Sci. Technol.* **50**, 4895–4904, doi:10.1021/acs.est.5b03827 (2016).
- 35 Silva Raquel, A., Adelman, Z., Fry Meridith, M. & West, J. J. The Impact of Individual Anthropogenic Emissions Sectors on the Global Burden of Human Mortality due to Ambient Air Pollution. *Environmental Health Perspectives* **124**, 1776–1784, doi:10.1289/EHP177 (2016).

- 36 Weagle, C. L. *et al.* Global Sources of Fine Particulate Matter: Interpretation of PM_{2.5} Chemical Composition Observed by SPARTAN using a Global Chemical Transport Model. *Environ. Sci. Technol.* **52**, 11670-11681, doi:10.1021/acs.est.8b01658 (2018).
- 37 Lee, C. J. *et al.* Response of Global Particulate-Matter-Related Mortality to Changes in Local Precursor Emissions. *Environ. Sci. Technol.* **49**, 4335-4344, doi:10.1021/acs.est.5b00873 (2015).
- 38 GBD 2017 Risk Factor Collaborators. Global, regional, and national comparative risk assessment of 84 behavioural, environmental and occupational, and metabolic risks or clusters of risks for 195 countries and territories, 1990-2017: a systematic analysis for the Global Burden of Disease Study 2017. *The Lancet* **392**, 1923-1994, doi:10.1016/S0140-6736(18)32225-6 (2018).
- 39 Grieshop, A. P., Marshall, J. D. & Kandlikar, M. Health and climate benefits of cookstove replacement options. *Energy Policy* **39**, 7530-7542, doi:<https://doi.org/10.1016/j.enpol.2011.03.024> (2011).

Reviewer comments, second round

Reviewer #1 (Remarks to the Author):

The authors have nicely addressed the issues raised, the paper may be accepted for publication

Reviewer #2 (Remarks to the Author):

My comments are now fully addressed. I would highly recommend the draft to be accept in current version.

Reviewer #3 (Remarks to the Author):

I would like to thank the authors for a very diligent, comprehensive and easily accessible response to the review comments. I have no further questions or suggestions and look forward to seeing this manuscript published.

Stefan Reis

Reviewer #4 (Remarks to the Author):

I was supportive of the original version of this paper and made only a few minor comments. The authors have responded appropriately to these.

The other reviewers made more detailed comments, although these others reviewers also appeared strongly supportive of the work presented. I have read through the comprehensive responses the authors have made to these other comments and suggestions. It is my view that the reviewers have provided appropriate responses and alterations and additions to the main paper and the supplementary information, and that the paper has been further improved as a result.

Response to Reviewers

We thank the Reviewers for the time spent re-reviewing our manuscript and for their positive comments.

REVIEWERS' COMMENTS

Reviewer #1 (Remarks to the Author):

The authors have nicely addressed the issues raised, the paper may be accepted for publication

Thank you to Reviewer #1 for their original comments and support of our latest version.

Reviewer #2 (Remarks to the Author):

My comments are now fully addressed. I would highly recommend the draft to be accept in current version.

Thank you to Reviewer #2 for their original comments and support of our latest version.

Reviewer #3 (Remarks to the Author):

I would like to thank the authors for a very diligent, comprehensive and easily accessible response to the review comments. I have no further questions or suggestions and look forward to seeing this manuscript published.

Stefan Reis

Thank you to Reviewer #3 for their original comments and support of our latest version.

Reviewer #4 (Remarks to the Author):

I was supportive of the original version of this paper and made only a few minor comments. The authors have responded appropriately to these.

The other reviewers made more detailed comments, although these others reviewers also appeared strongly supportive of the work presented. I have read through the comprehensive responses the authors have made to these other comments and suggestions. It is my view that the reviewers have provided appropriate responses and alterations and additions to the main paper and the supplementary information, and that the paper has been further improved as a result.

Thank you to Reviewer #4 for their original comments and support of our latest version.